# Out-of-distribution Representation Learning for Time Series Classification

**Wang Lu**[1]*,**Jindong Wang**[2]†, **Xinwei Sun**[3], **Yiqiang Chen**[1], **Xing Xie**[2]
[1]Institute of Comput. Tech., CAS   [2]Microsoft Research Asia   [3]Fudan University

## Abstract

Time series classification is an important problem in the real world. Due to its non-stationary property that the distribution changes over time, it remains challenging to build models for generalization to unseen distributions. In this paper, we propose to view time series classification from the *distribution* perspective. We argue that the temporal complexity of a time series dataset could attribute to unknown *latent* distributions that need characterize. To this end, we propose DIVERSIFY for out-of-distribution (OOD) representation learning on dynamic distributions of times series. DIVERSIFY takes an iterative process: it first obtains the *'worst-case'* latent distribution scenario via adversarial training, then reduces the gap between these latent distributions. We then show that such an algorithm is theoretically supported. Extensive experiments are conducted on seven datasets with different OOD settings across gesture recognition, speech commands recognition, wearable stress and affect detection, and sensor-based human activity recognition. Qualitative and quantitative results demonstrate that DIVERSIFY significantly outperforms other baselines and effectively characterizes the latent distributions. Code is available at `https://github.com/microsoft/robustlearn`.

## 1 Introduction

Time series classification is one of the most challenging problems in the machine learning and statistics community (Fawaz et al., 2019; Du et al., 2021). One important nature of time series is the non-stationary property, indicating that its statistical features are changing over time. For years, there have been tremendous efforts for time series classification, such as hidden Markov models (Fulcher & Jones, 2014), RNN-based methods (Hüsken & Stagge, 2003), and Transformer-based approaches (Li et al., 2019; Drouin et al., 2022).

We propose to model time series from the *distribution* perspective to handle its dynamically changing distributions; more precisely, to learn *out-of-distribution (OOD)* representations for time series that generalize to *unseen* distributions. The general OOD/domain generalization problem has been extensively studied (Wang et al., 2022; Lu et al., 2022; Krueger et al., 2021; Rame et al., 2022), where the key is to bridge the gap between known and unknown distributions. Despite existing efforts, OOD in time series remains *less studied* and more challenging. Compared to image classification, the *dynamic* distribution of time series data keeps changing over time, containing *diverse* distribution information that should be harnessed for better generalization.

Figure 1 shows an illustrative example. OOD generalization in image classification often involves several domains whose domain labels are static and known (subfigure (a)), which can be employed to build OOD models. However, Figure 1 (b) shows that in EMG time series data (Lobov et al., 2018), the distribution is changing dynamically over time and its domain information is *unavailable*. If no attention is paid to exploring its *latent* distributions (i.e., sub-domains), predictions may fail in face of diverse sub-domain distributions (subfigure (c)). This will dramatically impede existing OOD algorithms due to their reliance on domain information.

In this work, we propose DIVERSIFY, an OOD representation learning algorithm for time series classification by characterizing the latent distributions inside the data. Concretely speaking, DI-

---

*Work done when Wang Lu (luwang@ict.ac.cn) was an intern at Microsoft Research Asia.
†Correspondence to: Jindong Wang (jindong.wang@microsoft.com).

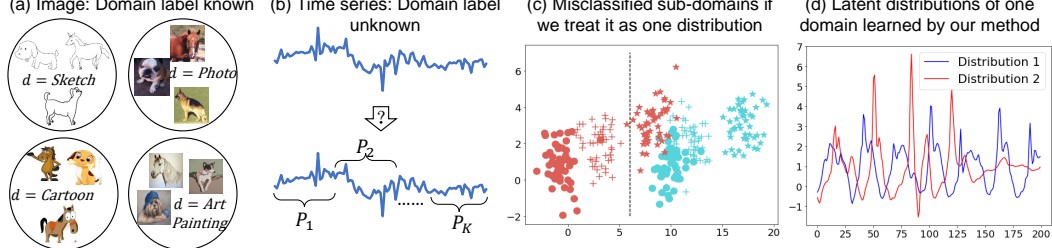

Figure 1: Illustration of DIVERSIFY: (a) Domain generalization for image data requires known domain labels. (b) Domain labels are unknown for time series. (c) If we treat the time series data as one single domain, the sub-domains are misclassified. Different colors and shapes correspond to different classes and domains. Axes represent data values. (d) Finally, our DIVERSIFY can effectively learn the latent distributions. X-axis represents data numbers while Y-axis represents values.

VERSIFY consists of a min-max adversarial game: on one hand, it learns to segment the time series data into several latent sub-domains by maximizing the segment-wise distribution gap to preserve diversities, i.e., the *'worst-case'* distribution scenario; on the other hand, it learns domain-invariant representations by reducing the distribution divergence between the obtained latent domains. Such latent distributions naturally exist in time series, e.g., the activity data from multiple people follow different distributions. Additionally, our experiments show that even the data of one person still has such diversity: it can also be split into several latent distributions. Figure 1 (d) shows that DIVERSIFY can effectively characterize the latent distributions (more results are in Sec. 3.5).

To summarize, our contributions are four-fold:

- **Novel perspective:** We propose to view time series classification from the distribution perspective to learn OOD representation, which is more challenging than the traditional image classification due to the existence of unidentified latent distributions.
- **Novel methodology:** DIVERSIFY is a novel framework to identify the latent distributions and learn generalized representations. Technically, we propose pseudo domain-class labels and adversarial self-supervised pseudo labeling to obtain the pseudo domain labels.
- **Theoretical insights:** We provide the theoretical insights behind DIVERSIFY to analyze its design philosophy and conduct experiments to prove the insights.
- **Superior performance and insightful results:** Qualitative and quantitative results using various backbones demonstrate the superiority of DIVERSIFY in several challenging scenarios: difficult tasks, significantly diverse datasets, and limited data. More importantly, DIVERSIFY can successfully characterize the latent distributions within a time series dataset.

## 2 METHODOLOGY

A time-series training dataset $\mathcal{D}^{tr}$ can be often pre-processed using sliding window[1] to $N$ inputs: $\mathcal{D}^{tr} = \{(\mathbf{x}_i, y_i)\}_{i=1}^{N}$, where $\mathbf{x}_i \in \mathcal{X} \subset \mathbb{R}^p$ is the $p$-dimensional instance and $y_i \in \mathcal{Y} = \{1, \ldots, C\}$ is its label. We use $\mathbb{P}^{tr}(\mathbf{x}, y)$ on $\mathcal{X} \times \mathcal{Y}$ to denote the joint distribution of the training dataset. Our goal is to learn a generalized model from $\mathcal{D}^{tr}$ to predict well on an *unseen* target dataset, $\mathcal{D}^{te}$, which is inaccessible in training. In our problem, the training and test datasets have the same input and output spaces but different distributions, i.e., $\mathcal{X}^{tr} = \mathcal{X}^{te}, \mathcal{Y}^{tr} = \mathcal{Y}^{te}$, but $\mathbb{P}^{tr}(\mathbf{x}, y) \neq \mathbb{P}^{te}(\mathbf{x}, y)$. We aim to train a model $h$ from $\mathcal{D}^{tr}$ to achieve minimum error on $\mathcal{D}^{te}$.

### 2.1 MOTIVATION

**What are domain and distribution shift in time series?** Time series may consist of several unknown latent distributions (domains), even if the dataset is fully labeled. For instance, data collected by sensors of three persons may belong to two different distributions due to their dissimilarities. This can be termed as spatial distribution shift. Surprisingly, we even find temporal distribution shifts in

---

[1]Sliding window is a common technique to segment one time series data into *fixed-size* windows. Each window is a *minimum* instance. We focus on fixed-size inputs for its popularity in time series (Das et al., 1998).

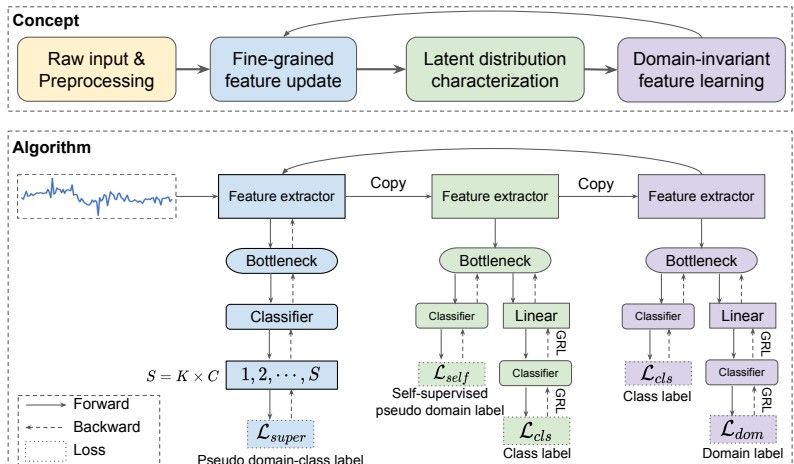

Figure 2: The framework of DIVERSIFY.

experiments (Figure 6) that distributions of one person can also change at different time. Those shifts widely exist in time series, as suggested by (Zhang et al., 2021; Ragab et al., 2022).

**OOD generalization requires latent domain characterization.** Due to the non-stationary property, naive approaches that treat time series as *one* distribution fail to capture domain-invariant (OOD) features since they ignore the diversities inside the dataset. In Figure 1 (c), we assume the training domain contains two sub-domains (circle and plus points). Directly treating it as one distribution via existing OOD approaches may generate the black margin. Red star points are misclassified to the green class when predicting on the OOD domain (star points) with the learned model. Thus, multiple diverse latent distributions in time series should be characterized to learn better OOD features.

**A brief formulation of latent domain characterization.** Following above discussions, a time series may consist of $K$ *unknown* latent domains[2][3] rather than a fixed one, i.e., $\mathbb{P}^{tr}(\mathbf{x}, y) = \sum_{i=1}^{K} \pi_i \mathbb{P}^i(\mathbf{x}, y)$, where $\mathbb{P}^i(\mathbf{x}, y)$ is the distribution of the $i$-th latent one with weight $\pi_i$, $\sum_{i=1}^{K} \pi_i = 1$.[4] There could be infinite ways to obtain $\mathbb{P}^i$s and our goal is to learn the *'worst-case'* distribution scenario where the distribution divergence between each $\mathbb{P}^i$ and $\mathbb{P}^j$ is maximized. Why the 'worst-case' scenario? It will maximally preserve the diverse information of each latent distribution, thus benefiting generalization. For an illustration, these obtained latent distributions are shown in Sec. 3.5.

## 2.2 DIVERSIFY

In this paper, we propose DIVERSIFY to learn OOD representations for time series classification. The core of DIVERSIFY is to characterize the latent distributions and then minimize the distribution divergence between each two. DIVERSIFY utilizes an iterative process: it first obtains the 'worst-case' distribution scenario from a given dataset, then bridges the distribution gaps between each pair of latent distributions. Figure 2 describes its main procedures, where steps $2 \sim 4$ are iterative:

1. Pre-processing: this step adopts the sliding window to split the entire training dataset into fixed-size windows. We argue that the data from one window is the smallest domain unit.

2. Fine-grained feature update: this step updates the feature extractor using the proposed *pseudo domain-class* labels as the supervision.

3. Latent distribution characterization: it aims to identify the domain label for each instance to obtain the latent distribution information. It *maximizes* the different distribution gaps to enlarge diversity.

4. Domain-invariant representation learning: this step utilizes pseudo domain labels from the last step to learn domain-invariant representations and train a generalizable model.

---

[2]We assume $K \in (1, N)$ as a smaller $K$ may be too coarse to show the diversities in distributions while a larger $K$ brings difficulties to optimization. $K$ is *tuned* in this work but we expect to learn it in the future.

[3]A domain is a set of data samples following a certain distribution and we use them interchangeably hereafter.

[4]We use the notations $\pi_i$ and $\mathbb{P}^i$ to only describe the problem, but do not formalize it.

**Fine-grained Feature Update.** Before characterizing the latent distributions, we perform fine-grained feature updates to obtain fine-grained representation. As shown in Figure 2 (blue), we propose a new concept: *pseudo domain-class label* to fully utilize the knowledge contained in domains and classes, which serves as the supervision for feature extractor. Features are more fine-grained w.r.t. domains *and* labels, instead of only attached to domains or labels.

At the first iteration, there is no domain label $d'$ and we simply initialize $d' = 0$ for all samples. We treat per category per domain as a *new* class with label $s \in \{1, 2, \cdots, S\}$. We have $S = K \times C$ where $K$ is the pre-defined number of latent distributions that can be tuned in experiments. We perform pseudo domain-class label assignment to get discrete values for supervision: $s = d' \times C + y$.

Let $h_f^{(2)}, h_b^{(2)}, h_c^{(2)}$ be feature extractor, bottleneck, and classifier, respectively (we use superscripts to denote step number). Then, the supervised loss is computed using the cross-entropy loss $\ell$:

$$\mathcal{L}_{super} = \mathbb{E}_{(\mathbf{x},y) \sim \mathbb{P}^{tr}} \ell \left( h_c^{(2)}(h_b^{(2)}(h_f^{(2)}(\mathbf{x}))), s \right). \tag{1}$$

**Latent Distribution Characterization.** This step characterizes the latent distributions contained in one dataset. As shown in Figure 2 (green), we propose an adapted version of adversarial training to disentangle the domain labels from the class labels. However, there are no actual domain labels provided, which hinders such disentanglement. Inspired by (Caron et al., 2018), we employ a *self-supervised pseudo-labeling* strategy to obtain domain labels.

First, we attain the centroid for each domain with class-invariant features:

$$\tilde{\mu}_k = \frac{\sum_{\mathbf{x}_i \in \mathcal{X}^{tr}} \delta_k(h_c^{(3)}(h_b^{(3)}(h_f^{(3)}(\mathbf{x}_i)))) h_b^{(3)}(h_f^{(3)}(\mathbf{x}_i))}{\sum_{\mathbf{x}_i \in \mathcal{X}^{tr}} \delta_k(h_c^{(3)}(h_b^{(3)}(h_f^{(3)}(\mathbf{x}_i))))}, \tag{2}$$

where $h_f^{(3)}, h_b^{(3)}, h_c^{(3)}$ are feature extractor, bottleneck, and classifier, respectively. $\tilde{\mu}_k$ is the initial centroid of the $k^{th}$ latent domain while $\delta_k$ is the $k^{th}$ element of the logit soft-max output. Then, we obtain the pseudo domain labels via the nearest centroid classifier using a distance function $D$:

$$\tilde{d}_i' = \arg\min_k D(h_b^{(3)}(h_f^{(3)}(\mathbf{x}_i)), \tilde{\mu}_k). \tag{3}$$

Then, we compute the centroids and obtain the updated pseudo domain labels:

$$\mu_k = \frac{\sum_{\mathbf{x}_i \in \mathcal{X}^{tr}} \mathbb{I}(\tilde{d}_i' = k) h_b^{(3)}(h_f^{(3)}(\mathbf{x}))}{\sum_{\mathbf{x}_i \in \mathcal{X}^{tr}} \mathbb{I}(\tilde{d}_i' = k)}, d_i' = \arg\min_k D(h_b^{(3)}(h_f^{(3)}(\mathbf{x}_i)), \mu_k), \tag{4}$$

where $\mathbb{I}(a) = 1$ when $a$ is true, otherwise 0. After obtaining $d'$, we can compute the loss of step 2:

$$\mathcal{L}_{self} + \mathcal{L}_{cls} = \mathbb{E}_{(\mathbf{x},y) \sim \mathbb{P}^{tr}} \ell(h_c^{(3)}(h_b^{(3)}(h_f^{(3)}(\mathbf{x}))), d') + \ell(h_{adv}^{(3)}(R_{\lambda_1}(h_b^{(3)}(h_f^{(3)}(\mathbf{x})))), y), \tag{5}$$

where $h_{adv}^{(3)}$ is the discriminator for step 3 that contains several linear layers and one classification layer. $R_{\lambda_1}$ is the gradient reverse layer with hyperparameter $\lambda_1$ (Ganin et al., 2016). After this step, we can obtain pseudo domain label $d'$ for $\mathbf{x}$.

**Domain-invariant Representation Learning.** After obtaining the latent distributions, we learn domain-invariant representations for generalization. In fact, this step (purple in Figure 2) is simple: we borrow the idea from DANN (Ganin et al., 2016) and directly use adversarial training to update the classification loss $\mathcal{L}_{cls}$ and domain classifier loss $\mathcal{L}_{dom}$ using gradient reversal layer (GRL) (a common technique that facilitates adversarial training via reversing gradients) (Ganin et al., 2016):

$$\mathcal{L}_{cls} + \mathcal{L}_{dom} = \mathbb{E}_{(\mathbf{x},y) \sim \mathbb{P}^{tr}} \ell(h_c^{(4)}(h_b^{(4)}(h_f^{(4)}(\mathbf{x}))), y) + \ell(h_{adv}^{(4)}(R_{\lambda_2}(h_b^{(4)}(h_f^{(4)}(\mathbf{x})))), d'), \tag{6}$$

where $\ell$ is the cross-entropy loss and $R_{\lambda_2}$ is the gradient reverse layer with hyperparameter $\lambda_2$ (Ganin et al., 2016). We will omit the details of GRL and adversarial training here since they are common techniques in deep learning. More details are presented in Appendix B.2.

**Training, Inference, and Complexity.** We repeat these steps until convergence or max epochs. Different from existing methods, the last two steps only optimize the last few independent layers but not the feature extractor. We perform inference with the modules from the last step. Most of the trainable parameters are shared between modules, indicating that DIVERSIFY has the same model size as existing methods and can reach quick convergence in experiments (Figure F.5).

## 2.3 THEORETICAL INSIGHTS

We present some theoretical insights to show that our approach is well motivated in theory. Proofs can be found in Appendix A.

**Proposition 2.1.** *Let $\mathcal{X}$ be a space and $\mathcal{H}$ be a class of hypotheses corresponding to this space. Let $\mathbb{Q}$ and the collection $\{\mathbb{P}_i\}_{i=1}^{K}$ be distributions over $\mathcal{X}$ and let $\{\varphi_i\}_{i=1}^{K}$ be a collection of non-negative coefficient with $\sum_i \varphi_i = 1$. Let $\mathcal{O}$ be a set of distributions s.t. $\forall \mathbb{S} \in \mathcal{O}$, the following holds*

$$d_{\mathcal{H}\Delta\mathcal{H}}(\sum_i \varphi_i \mathbb{P}_i, \mathbb{S}) \le \max_{i,j} d_{\mathcal{H}\Delta\mathcal{H}}(\mathbb{P}_i, \mathbb{P}_j). \tag{7}$$

*Then, for any $h \in \mathcal{H}$,*

$$\varepsilon_{\mathbb{Q}}(h) \le \lambda' + \sum_i \varphi_i \varepsilon_{\mathbb{P}_i}(h) + \frac{1}{2} \min_{\mathbb{S}\in\mathcal{O}} d_{\mathcal{H}\Delta\mathcal{H}}(\mathbb{S}, \mathbb{Q}) + \frac{1}{2} \max_{i,j} d_{\mathcal{H}\Delta\mathcal{H}}(\mathbb{P}_i, \mathbb{P}_j), \tag{8}$$

*where $\lambda'$ is the error of an ideal joint hypothesis. $\varepsilon_{\mathbb{P}}(h)$ is the error for a hypothesis $h$ on a distribution $\mathbb{P}$. $d_{\mathcal{H}\Delta\mathcal{H}}(\mathbb{P}, \mathbb{Q})$ is $\mathcal{H}$-divergence which measures differences in distribution (Ben-David et al., 2010).*

The first item in Eq. (8), $\lambda'$, is often neglected since it is small in reality. The second item, $\sum_i \varphi_i \varepsilon_{\mathbb{P}_i}(h)$, exists in almost all methods and can be minimized via supervision from class labels with cross-entropy loss in Eq. (6). Our main purpose is to minimize the last two items in Eq. (8). Here $\mathbb{Q}$ corresponds to the unseen out-of-distribution target domain.

The last term $\frac{1}{2} \max_{i,j} d_{\mathcal{H}\Delta\mathcal{H}}(\mathbb{P}_i, \mathbb{P}_j)$ is common in OOD theory which measures the maximum differences among source domains. This corresponds to step 4 in our approach.

Finally, the third item, $\frac{1}{2} \min_{\mathbb{S}\in\mathcal{O}} d_{\mathcal{H}\Delta\mathcal{H}}(\mathbb{S}, \mathbb{Q})$, explains why we exploit sub-domains in step 3. Since our goal is to learn a model which can perform well on an unseen target domain, we cannot obtain $\mathbb{Q}$. To minimize $\frac{1}{2} \min_{\mathbb{S}\in\mathcal{O}} d_{\mathcal{H}\Delta\mathcal{H}}(\mathbb{S}, \mathbb{Q})$, we can only enlarge the range of $\mathcal{O}$. We have to $\max_{i,j} d_{\mathcal{H}\Delta\mathcal{H}}(\mathbb{P}_i, \mathbb{P}_j)$ according to Eq. (7), corresponding to step 3 in our method which tries to segment the time series data into several latent sub-domains by maximizing the segment-wise distribution gap to preserve diversities, i.e., the 'worst-case' distribution scenario.

## 3 EXPERIMENTS

We perform evaluations on four diverse time series classification tasks: gesture recognition, speech commands recognition, wearable stress&affect detection, and sensor-based activity recognition.

Time series OOD algorithms are currently less studied and there are only two recent strong approaches for comparison: GILE (Qian et al., 2021) and AdaRNN (Du et al., 2021).[5] We further compare with 7 general OOD methods[6] from DomainBed (Gulrajani & Lopez-Paz, 2021): ERM, DANN (Ganin et al., 2016), CORAL (Sun & Saenko, 2016), Mixup (Zhang et al., 2018), GroupDRO (Sagawa et al., 2020), RSC (Huang et al., 2020), and ANDMask (Parascandolo et al., 2021). **More details of these methods are in Sec. B.2 and B.3.** For fairness, all methods (except GILE and AdaRNN) use a feature net with two blocks and each block has one convolution layer, one pooling layer, and one batch normalization layer, following (Wang et al., 2019). We also use **Transformers** (Vaswani et al., 2017) for backbone. **Detailed data pre-processing, architecture, and hyperparameters are in Appendix C.5 and D.** Ablations with various backbones are in Figure 8 and Appendix F.4.

Most OOD methods require the domain labels known in training while ours does not, which is more challenging and practical. We conduct the training-domain-validation strategy and the training data are split by $8 : 2$ for training and validation. We tune all methods to report the average best performance of three trials for fairness. Note that the "target" in experiments is unseen and only used for testing. $K$ in DIVERSIFY is treated as a hyperparameter and we tune it to record the best OOD performance.[7] Per-segment accuracy is the evaluation metric. **Time complexity and convergence are in Sec. F.5**, showing its quick convergence.

---

[5] There are recent approaches purely on time series, but not for OOD.

[6] There could be recent OOD methods, but according to DomainBed (Gulrajani & Lopez-Paz, 2021), most approaches do not significantly outperform ERM. DANN, CORAL, and Mixup are also strong baselines.

[7] There might be no optimal $K$ for a dataset. We perform grid search in $[2, 10]$ to get the best performance.

## 3.1 GESTURE RECOGNITION

First, we evaluate DIVERSIFY on EMG for gestures Data Set (Lobov et al., 2018). It contains data of 36 subjects with 7 classes and we select 6 common classes for our experiments. We randomly divide 36 subjects into four domains (i.e., 0, 1, 2, 3). More details on EMG and domain splits can be found in Sec. C.2 and C.6 respectively. EMG data is affected by many factors since it comes from bioelectric signals. EMG data are scene and device-dependent, which means the same person may generate different data when performing the same activity with the same device at a different time (i.e., distribution shift across time (Wilson et al., 2020; Purushotham et al., 2016)) or with the different devices at the same time. Thus, the EMG benchmark is challenging. Table 1 shows that with the same

Table 1: Results on EMG dataset. "Target" $0 \sim 4$ denotes unseen test distribution that is only for testing.

| Target | 0 | 1 | 2 | 3 | AVG |
|---|---|---|---|---|---|
| ERM | 62.6 | 69.9 | 67.9 | 69.3 | 67.4 |
| DANN | 62.9 | 70.0 | 66.5 | 68.2 | 66.9 |
| CORAL | 66.4 | 74.6 | 71.4 | 74.2 | 71.7 |
| Mixup | 60.7 | 69.9 | 70.5 | 68.2 | 67.3 |
| GroupDRO | 67.6 | 77.4 | 73.7 | 72.5 | 72.8 |
| RSC | 70.1 | 74.6 | 72.4 | 71.9 | 72.2 |
| ANDMask | 66.5 | 69.1 | 71.4 | 68.9 | 69.0 |
| AdaRNN | 68.8 | 81.1 | 75.3 | **78.1** | 75.8 |
| DIVERSIFY | **71.7** | **82.4** | **76.9** | 77.3 | **77.1** |

backbone, our method achieves the best average performance and is **4.3**% better than the second-best method. DIVERSIFY even outperforms AdaRNN which has a stronger backbone.

## 3.2 SPEECH COMMANDS

Then, we adopt a regular speech recognition task, the Speech Commands dataset (Warden, 2018). It consists of one-second audio recordings of both background noise and spoken words such as 'left' and 'right'. It is collected from more than 2,000 persons, thus is more complicated. Following (Kidger et al., 2020), we use 34,975 time series corresponding to ten spoken words to produce a balanced classification problem. Since this dataset is collected from multiple persons, the training and test distributions are different, which is also an OOD problem with one training domain. There are many subjects and each subject only records a few audios. Thus, we do not split each sample. Figure 3 shows the results on two different backbones. Compared with GroupDRO, DIVERSIFY has over **1**% improvement with a basic CNN backbone and over **0.6**% improvement with a

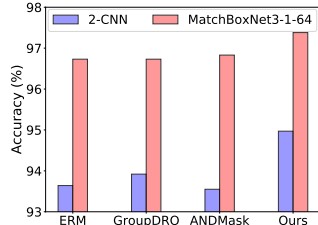

Figure 3: Results on Speech commands with two different backbones.

strong backbone MatchBoxNet3-1-64 (Majumdar & Ginsburg, 2020). It demonstrates the superiority of our method on a regular time-series benchmark containing massive distributions.

## 3.3 WEARABLE STRESS AND AFFECT DETECTION

We further evaluate DIVERSIFY on a larger dataset, Wearable Stress and Affect Detection (WESAD) (Schmidt et al., 2018). WESAD is a public dataset that contains physiological and motion data of 15 subjects with $63,000,000$ instances. We utilize sensor modalities of chest-worn devices including electrocardiogram, electrodermal activity, electromyogram, respiration, body temperature, and three axis acceleration. We split 15 subjects into four domains (details are in Sec. C.6). Results Figure 4 showed that our method achieves the best performance compared to other state-of-the-art methods with an improvement of over **8**% on this larger dataset.

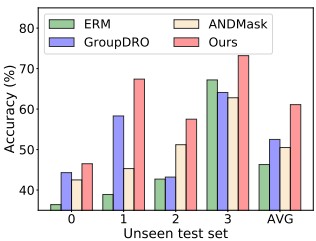

Figure 4: Results on WESAD. Here, $0 \sim 4$ in x-axis denotes the unseen test dataset.

## 3.4 SENSOR-BASED HUMAN ACTIVITY RECOGNITION

Finally, we construct *four* diverse OOD settings by leveraging four sensor-based human activity recognition datasets: DSADS (Barshan & Yüksek, 2014), USC-HAD (Zhang & Sawchuk, 2012), UCI-HAR (Anguita et al., 2012), and PAMAP (Reiss & Stricker, 2012). These datasets are collected from different people and positions using accelerometer and gyroscope, with $11,741,000$ instances in total. (1) **Cross-person generalization** aims to learn generalized models for different persons. (2) **Cross-position generalization** aims to learn generalized models for different sensor positions. (3) **Cross-dataset generalization** aims to learn generalized models for different datasets. (4) **One-Person-To-Another** aims to learn generalized models for different persons from data of a single

Table 2: Accuracy on cross-person generalization. "Target" $0 \sim 4$ denotes the unseen test set.

| | DSADS | | | | | USC-HAD | | | | | PAMAP | | | | | ALL |
|---|---|---|---|---|---|---|---|---|---|---|---|---|---|---|---|---|
| Target | 0 | 1 | 2 | 3 | AVG | 0 | 1 | 2 | 3 | AVG | 0 | 1 | 2 | 3 | AVG | AVG |
| ERM | 83.1 | 79.3 | 87.8 | 71.0 | 80.3 | 81.0 | 57.7 | 74.0 | 65.9 | 69.7 | 90.0 | 78.1 | 55.8 | 84.4 | 77.1 | 75.7 |
| DANN | 89.1 | 84.2 | 85.9 | 83.4 | 85.6 | 81.2 | 57.9 | 76.7 | 70.7 | 71.6 | 82.2 | 78.1 | 55.4 | 87.3 | 75.7 | 77.7 |
| CORAL | 91.0 | 85.8 | 86.6 | 78.2 | 85.4 | 78.8 | 58.9 | 75.0 | 53.7 | 66.6 | 86.2 | 77.8 | 49.0 | 87.8 | 75.2 | 75.7 |
| Mixup | 89.6 | 82.2 | 89.2 | 86.9 | 87.0 | 80.0 | 64.1 | 74.3 | 61.3 | 69.9 | 89.4 | 80.3 | 58.4 | 87.7 | 79.0 | 78.6 |
| GroupDRO | 91.7 | 85.9 | 87.6 | 78.3 | 85.9 | 80.1 | 55.5 | 74.7 | 60.0 | 67.6 | 85.2 | 77.7 | 56.2 | 85.0 | 76.0 | 76.5 |
| RSC | 84.9 | 82.3 | 86.7 | 77.7 | 82.9 | 81.9 | 57.9 | 73.4 | 65.1 | 69.6 | 87.1 | 76.9 | 60.3 | 87.8 | 78.0 | 76.9 |
| ANDMask | 85.0 | 75.8 | 87.0 | 77.6 | 81.4 | 79.9 | 55.3 | 74.5 | 65.0 | 68.7 | 86.7 | 76.4 | 43.6 | 85.6 | 73.1 | 74.4 |
| GILE | 81.0 | 75.0 | 77.0 | 66.0 | 74.7 | 78.0 | 62.0 | 77.0 | 63.0 | 70.0 | 83.0 | 68.0 | 42.0 | 76.0 | 67.5 | 70.7 |
| AdaRNN | 80.9 | 75.5 | 90.2 | 75.5 | 80.5 | 78.6 | 55.3 | 66.9 | 73.7 | 68.6 | 81.6 | 71.8 | 45.4 | 82.7 | 70.4 | 73.2 |
| DIVERSIFY | 90.4 | 86.5 | 90.0 | 86.1 | 88.2 | 82.6 | 63.5 | 78.7 | 71.3 | 74.0 | 91.0 | 84.3 | 60.5 | 87.7 | 80.8 | 81.0 |

Table 3: Classification accuracy on cross-position, cross-dataset, and one-to-another generalization.

| | Cross-position generalization | | | | | | Cross-dataset generalization | | | | | One-Person-To-Another | | | |
|---|---|---|---|---|---|---|---|---|---|---|---|---|---|---|---|
| Target | 0 | 1 | 2 | 3 | 4 | AVG | 0 | 1 | 2 | 3 | AVG | DSADS | USC-HAD | PAMAP | AVG |
| ERM | 41.5 | 26.7 | 35.8 | 21.4 | 27.3 | 30.6 | 26.4 | 29.6 | 44.4 | 32.9 | 33.3 | 51.3 | 46.2 | 53.1 | 50.2 |
| DANN | 45.4 | 25.3 | 38.1 | 28.9 | 25.1 | 32.6 | 29.7 | 45.3 | 46.1 | 43.8 | 41.2 | - | - | - | - |
| CORAL | 33.2 | 25.2 | 25.8 | 22.3 | 20.6 | 25.4 | 39.5 | 41.8 | 39.1 | 36.6 | 39.2 | - | - | - | - |
| Mixup | 48.8 | 34.2 | 37.5 | 29.5 | 29.9 | 36.0 | 37.3 | 47.4 | 40.2 | 23.1 | 37.0 | 62.7 | 46.3 | 58.6 | 55.8 |
| GroupDRO | 27.1 | 26.7 | 24.3 | 18.4 | 24.8 | 24.3 | 51.4 | 36.7 | 33.2 | 33.8 | 38.8 | 51.3 | 48.0 | 53.1 | 50.8 |
| RSC | 46.6 | 27.4 | 35.9 | 27.0 | 29.8 | 33.3 | 33.1 | 39.7 | 45.3 | 45.9 | 41.0 | 59.1 | 49.0 | 59.7 | 55.9 |
| ANDMask | 47.5 | 31.1 | 39.2 | 30.2 | 29.9 | 35.6 | 41.7 | 33.8 | 43.2 | 40.2 | 39.7 | 57.2 | 45.9 | 54.3 | 52.5 |
| DIVERSIFY | 47.7 | 32.9 | 44.5 | 31.6 | 30.4 | 37.4 | 48.7 | 46.9 | 49.0 | 59.9 | 51.1 | 67.6 | 55.0 | 62.5 | 61.7 |

person.[8] For simplicity, we use $0, 1, \cdots$ to denote different domains. **More details on datasets information, setting construction, and domain splits can be found in Sec. C.3, C.4, and C.6.**

Table 2 and Table 3 show the results on four settings for HAR, where our method significantly outperforms the second-best baseline by $2.4\%$, $1.4\%$, $9.9\%$, and $5.8\%$ respectively. All results demonstrate the superiority of DIVERSIFY. More results using Transformer can be found in F.4.

We observe more insightful conclusions. (1) *When the task is difficult:* In the Cross-Person setting, USC-HAD may be the most difficult task. Although it has more samples, it contains 14 subjects with only two sensors on one position, which may bring more difficulty in learning. The results prove the above argument that all methods perform terribly on this benchmark while ours has the largest improvement. (2) *When datasets are significantly more diverse:* Compared to Cross-Person and Cross-Position settings, Cross-Dataset may be more difficult since all datasets are totally different and samples are influenced by subjects, devices, sensor positions, and some other factors. In this setting, our method is substantially better than others. (3) *Limited data:* Compared with Cross-Person setting, One-Person-To-Another is more difficult since it has fewer data samples. In this case, enhancing diversity can bring a remarkable improvement and our method can boost the performance.

## 3.5 ANALYSIS

**Ablation study** We present ablation study to answer the following three questions. (1) *Why obtaining pseudo domain labels with class-invariant features in step 3?* If we obtain pseudo domain labels with common features, domain labels may have correlations with class labels, which may introduce

---

[8] In One-Person-To-Another setting, we only report average accuracy of four tasks on each dataset. Since only one domain exists in training dataset for this setting, DANN and CORAL cannot be implemented here.

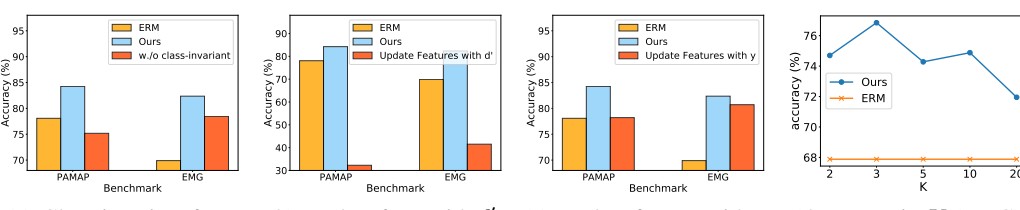

(a) Class-invariant feat.     (b) Update feat. with $d'$     (c) Update feature with $y$     (d) #Domain $K$ (EMG)

Figure 5: Ablation study of DIVERSIFY.

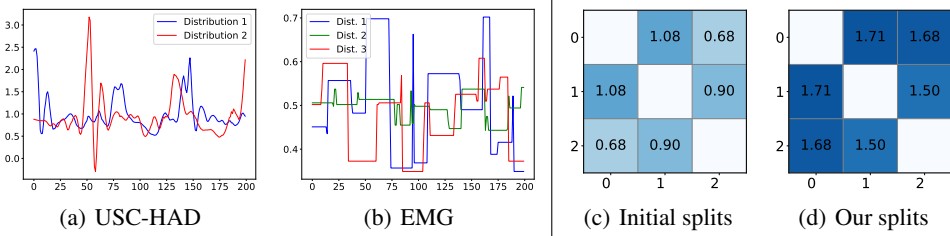

| (a) USC-HAD | (b) EMG | (c) Initial splits | (d) Our splits |

Figure 6: (a) (b) Latent distributions obtained by our method on two datasets. X-axis is data numbers while Y-axis is its values. (c) (d) $\mathcal{H}$-divergence among domains with initial splits and our splits on PAMAP. Axes are domain numbers.

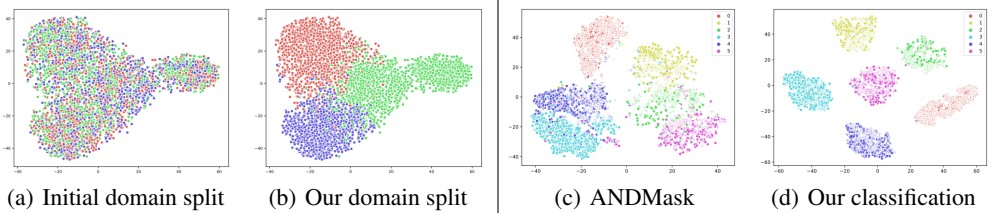

| (a) Initial domain split | (b) Our domain split | (c) ANDMask | (d) Our classification |

Figure 7: t-SNE visualizations for domain splits ((a) (b)) and classification ((c) (d)) on EMG data.

contradictions when learning domain-invariant representations and lead to common performance. This is certified by the results in Figure 5(a). (2) *Why using fine-grained domain-class labels in step 2?* If we utilize pseudo domain labels to update the feature net, it may make the representations seriously biased towards domain-related features and thereby leads to terrible performance on classification, which is proved in Figure 5(b). If we only utilize class labels to update the feature net, it may make representations biased to class-related features, thus DIVERSIFY is unable to obtain true latent sub-domains, as shown in Figure 5(c). Hence, we should employ fine-grained domain-class labels to obtain representations with both domain and class information. (3) *The more latent domains, the better?* More latent domains may not bring better results (Figure 5(d)) since a dataset may only have a few latent domains and introducing more may contradict its intrinsic data property. Plus, more latent domains also make it harder to obtain pseudo domain labels and learn domain-invariant features.

**Existence of latent distributions**   What exactly can our DIVERSIFY learn? In Figure 6(a), for a subject in USC-HAD, there is more than one latent distribution for his walking activities, showing the existence of temporal distribution shift: the distribution of the same activity could change. For spatial distribution shift, Figure 6(b) on EMG dataset shows that our algorithm found three latent distributions from the EMG data of multiple people. These results indicate the existence of latent distributions with both temporal and spatial distribution shifts. [9] More results are in Appendix F.1.

**Quantitative analysis for 'worst-case' distributions**   We present quantitative analysis by computing the $\mathcal{H}$-divergence (Ben-David et al., 2010) to show the effectiveness of our 'worst-case' distribution'. As shown in Figure 6(c) and 6(d), compared to initial domain splits, latent sub-domains generated by our method have larger $\mathcal{H}$-divergence among each other. According to Prop. 2.1, larger $\mathcal{H}$-divergence among domains brings better generalization. This again shows the efficacy of DIVERSIFY in computing the 'worst-case' distribution scenario. More results are in Appendix F.3.

**Visualization study**   We present some visualizations to show the rationales of DIVERSIFY. Data points with different initial domain labels are mixed together in Figure 7(a) while DIVERSIFY can characterize different latent distributions and separate them well in Figure 7(b). Figure 7(d) and 7(c) show that DIVERSIFY can learn better domain-invariant representations compared to the latest method ANDMask. To sum up, DIVERSIFY can find better latent domains to enhance generalization. More results are in Appendix F.1.

**Varying backbones**   Figure 8 shows the results using small, medium, and large backbones, respectively (we implement them with different numbers of layers.). Results indicate that larger models tend

---

[9]Figure 6(c)-6(d) and the experimental results above prove that paying attention to shifts comprehensively can bring larger divergence and better results.

to achieve better OOD generalization performance. Our method outperforms others in all backbones, showing that DIVERSIFY presents consistently strong OOD performance in different architectures. More results with Transformer are in Appendix F.4 Parameter sensitivity is in Appendix F.2.

## 4 RELATED WORK

**Time series classification** is a challenging problem. Researches mainly focus on temporal relation modeling via specially-designed methods (Dempster et al., 2021), RNN-based networks (Dennis et al., 2019), or Transformer architecture (Drouin et al., 2022). To our best knowledge, there is only one recent work (Du et al., 2021) that studied time series from the distribution level. However, AdaRNN is a two-stage non-differential method that is tailored for RNN.

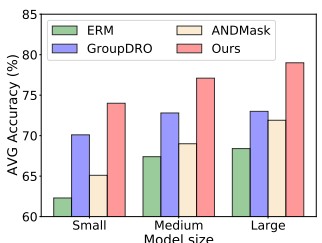

Figure 8: Results with different backbones on EMG.

**Domain / OOD generalization** (Wang et al., 2022; Lu et al., 2022) typically assumes the availability of domain labels for training. Specifically, Matsuura & Harada (2020) also studied DG without domain labels by clustering with the style features for images, which is not applied to time series and is not end-to-end trainable. Disentanglement (Peng et al., 2019; Zhang et al., 2022b) tries to disentangle the domain and label information, but they also assume access to domain information. **Single domain generalization** is similar to our problem setting which also involves one training domain (Fan et al., 2021; Li et al., 2021; Wang et al., 2021; Zhu & Li, 2022). However, they treated the single domain as *one* distribution and did not explore latent distributions.

**Multi-domain learning** is similar to DG which also trains on multiple domains, but also tests on training distributions. Deecke et al. (2022) proposed sparse latent adapters to learn from unknown domain labels, but their work does not consider the min-max worst-case distribution scenario and optimization. In **domain adaptation**, Wang et al. (2020) proposed the notion of domain index and further used variational models to learn them (Xu et al., 2023), but took a different modeling methodology since they did not consider min-max optimization. **Mixture models** (Rasmussen et al., 1999) are models representing the presence of subpopulations within an overall population, e.g., Gaussian mixture models. Our approach has a similar formulation but does not use generative models. Subpopulation shift is a new setting (Koh et al., 2021) that refers to the case where the training and test domains overlap, but their relative proportions differ. Our problem does not belong to this setting since we assume that these distributions do not overlap.

**Distributionally robust optimization** (Delage & Ye, 2010) shares a similar paradigm with our work, whose paradigm is also to seek a distribution that has the worst performance within a range of the raw distribution. GroupDRO (Sagawa et al., 2020) studied DRO at a group level. However, we study the internal distribution shift instead of seeking a global distribution close to the original one.

## 5 LIMITATION AND DISCUSSION

DIVERSIFY could be more perfect by pursuing the following avenues. 1) Estimate the number of latent distributions $K$ automatically: we currently treat it as a hyperparameter. 2) Seek the semantics behind latent distributions (e.g., Figure 6(a)): can adding more human knowledge obtain better latent distributions? 3) Extend DIVERSIFY beyond classification, but for forecasting problems.

Moreover, we argue that dynamic distributions not only exist in time series, but also in general machine learning data such as images and text (Deecke et al., 2022; Xu et al., 2023). Thus, it is of great interest to apply our approach to these domains to further improve their performance.

## 6 CONCLUSION

We proposed DIVERSIFY to learn generalized representation for time series classification. DIVERSIFY employs an adversarial game that maximizes the 'worst-case' distribution scenario while minimizing their distribution divergence. We demonstrated its effectiveness in different applications. We are surprised that not only a mixed dataset, but one dataset from a single person can also contain several latent distributions. Characterizing such latent distributions will greatly improve the generalization performance on unseen datasets.

## ACKNOWLEDGEMENT

This work is supported by National Key Research & Development Program of China (No. 2020YFC2007104), Natural Science Foundation of China (No. 61972383), the Strategic Priority Research Program of Chinese Academy of Sciences ( No. XDA28040500), Science Research Foundation of the Joint Laboratory Project on Digital Ophthalmology and Vision Science (No.SZYK202201).

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

CONTENTS

## A  THEORETICAL INSIGHTS

### A.1  BACKGROUND

For a distribution $\mathbb{P}$ with an ideal binary labeling function $h^*$ and a hypothesis $h$, we define the error $\varepsilon_{\mathbb{P}}(h)$ in accordance with (Ben-David et al., 2010) as:

$$\varepsilon_{\mathbb{P}}(h) = \mathbb{E}_{\mathbf{x} \sim \mathbb{P}} |h(\mathbf{x}) - h^*(\mathbf{x})|. \tag{9}$$

We also give the definition of $\mathcal{H}$-divergence according with (Ben-David et al., 2010). Given two distributions $\mathbb{P}, \mathbb{Q}$ over a space $\mathcal{X}$ and a hypothesis class $\mathcal{H}$,

$$d_{\mathcal{H}}(\mathbb{P}, \mathbb{Q}) = 2 \sup_{h \in \mathcal{H}} |Pr_{\mathbb{P}}(I_h) - Pr_{\mathbb{Q}}(I_h)|, \tag{10}$$

where $I_h = \{\mathbf{x} \in \mathcal{X} | h(\mathbf{x}) = 1\}$. We often consider the $\mathcal{H}\Delta\mathcal{H}$-divergence in (Ben-David et al., 2010) where the symmetric difference hypothesis class $\mathcal{H}\Delta\mathcal{H}$ is the set of functions characteristic to disagreements between hypotheses.

**Theorem A.1.** *(Theorem 2.1 in (Sicilia et al., 2021), modified from Theorem 2 in (Ben-David et al., 2010)). Let $\mathcal{X}$ be a space and $\mathcal{H}$ be a class of hypotheses corresponding to this space. Suppose $\mathbb{P}$ and $\mathbb{Q}$ are distributions over $\mathcal{X}$. Then for any $h \in \mathcal{H}$, the following holds*

$$\varepsilon_{\mathbb{Q}}(h) \leq \lambda'' + \varepsilon_{\mathbb{P}}(h) + \frac{1}{2}d_{\mathcal{H}\Delta\mathcal{H}}(\mathbb{Q}, \mathbb{P}) \tag{11}$$

*with $\lambda''$ the error of an ideal joint hypothesis for $\mathbb{Q}, \mathbb{P}$.*

Theorem A.1 provides an upper bound on the target-error. $\lambda''$ is a property of the dataset and hypothesis class and is often ignored. Theorem A.1 demonstrates the necessity to learn domain invariant features.

## A.2 PROOF OF PROPOSITION 2.1.

**Proposition 2.1.** Let $\mathcal{X}$ be a space and $\mathcal{H}$ be a class of hypotheses corresponding to this space. Let $\mathbb{Q}$ and the collection $\{\mathbb{P}_i\}_{i=1}^{K}$ be distributions over $\mathcal{X}$ and let $\{\varphi_i\}_{i=1}^{K}$ be a collection of non-negative coefficient with $\sum_i \varphi_i = 1$. Let the object $\mathcal{O}$ be a set of distributions such that for every $\mathbb{S} \in \mathcal{O}$ the following holds

$$d_{\mathcal{H}\Delta\mathcal{H}}(\sum_i \varphi_i\mathbb{P}_i, \mathbb{S}) \leq \max_{i,j} d_{\mathcal{H}\Delta\mathcal{H}}(\mathbb{P}_i, \mathbb{P}_j). \tag{12}$$

Then, for any $h \in \mathcal{H}$,

$$\varepsilon_{\mathbb{Q}}(h) \leq \lambda' + \sum_i \varphi_i\varepsilon_{\mathbb{P}_i}(h) + \frac{1}{2}\min_{\mathbb{S}\in\mathcal{O}} d_{\mathcal{H}\Delta\mathcal{H}}(\mathbb{S}, \mathbb{Q}) + \frac{1}{2}\max_{i,j} d_{\mathcal{H}\Delta\mathcal{H}}(\mathbb{P}_i, \mathbb{P}_j) \tag{13}$$

where $\lambda'$ is the error of an ideal joint hypothesis.

*Proof.* On one hand, with Theorem A.1, we have

$$\varepsilon_{\mathbb{Q}}(h) \leq \lambda_1' + \varepsilon_{\mathbb{S}}(h) + \frac{1}{2}d_{\mathcal{H}\Delta\mathcal{H}}(\mathbb{S}, \mathbb{Q}), \forall h \in \mathcal{H}, \forall \mathbb{S} \in \mathcal{O}. \tag{14}$$

On the other hand, with Theorem A.1, we have

$$\varepsilon_{\mathbb{S}}(h) \leq \lambda_2' + \varepsilon_{\sum_i \varphi_i\mathbb{P}_i}(h) + \frac{1}{2}d_{\mathcal{H}\Delta\mathcal{H}}(\sum_i \varphi_i\mathbb{P}_i, \mathbb{S}), \forall h \in \mathcal{H}. \tag{15}$$

Since $\varepsilon_{\sum_i \varphi_i\mathbb{P}_i}(h) = \sum_i \varphi_i\varepsilon_{\mathbb{P}_i}(h)$, and $d_{\mathcal{H}\Delta\mathcal{H}}(\sum_i \varphi_i\mathbb{P}_i, \mathbb{S}) \leq \max_{i,j} d_{\mathcal{H}\Delta\mathcal{H}}(\mathbb{P}_i, \mathbb{P}_j)$, we have

$$\varepsilon_{\mathbb{Q}}(h) \leq \lambda' + \sum_i \varphi_i\varepsilon_{\mathbb{P}_i}(h) + \frac{1}{2}d_{\mathcal{H}\Delta\mathcal{H}}(\mathbb{S}, \mathbb{Q}) + \frac{1}{2}\max_{i,j} d_{\mathcal{H}\Delta\mathcal{H}}(\sum_i \varphi_i\mathbb{P}_i, \mathbb{S}), \forall h \in \mathcal{H}, \forall \mathbb{S} \in \mathcal{O}, \tag{16}$$

where $\lambda' = \lambda_1' + \lambda_2'$. Equation 16 for all $\mathbb{S} \in \mathcal{O}$ holds. Therefore, we complete the proof. $\square$

## B METHOD DETAILS

### B.1 DOMAIN-INVARIANT REPRESENTATION LEARNING

Domain-invariant representation learning utilizes adversarial training which contains a feature network, a domain discriminator, and a classification network. The domain discriminator tries its best to discriminate domain labels of data while the feature network tries its best to generate features to confuse the domain discriminator, which thereby obtains domain-invariant representation. Therefore, it is an adversarial process, and in our setting, it can be expressed as follows,

$$\min_{h_b^{(4)}, h_c^{(4)}} \mathbb{E}_{(\mathbf{x},y)\sim\mathbb{P}^{tr}} \ell(h_c^{(4)}(h_b^{(4)}(h_f^{(4)}(\mathbf{x}))), y) - \ell(h_{adv}^{(4)}(h_b^{(4)}(h_f^{(4)}(\mathbf{x}))), d'),$$

$$\min_{h_{adv}^{(4)}} \mathbb{E}_{(\mathbf{x},y)\sim\mathbb{P}^{tr}} \ell(h_{adv}^{(4)}(h_b^{(4)}(h_f^{(4)}(\mathbf{x}))), d'). \tag{17}$$

To optimize Eq. (17), we need an iterative process to optimize $h_b^{(4)}$, $h_c^{(4)}$ and $h_{adv}^{(4)}$ iteratively, which is cumbersome. It is better to optimize $h_b^{(4)}$, $h_c^{(4)}$ and $h_{adv}^{(4)}$ at the same time. It is obvious that the key is to solve the problems caused by the negative sign in Eq. (17). Therefore, a special gradient reversal layer (GRL), a popular implementation of the adversarial training in training several domains as suggested by (Ganin et al., 2016), came. GRL acts as an identity transformation during the forward propagation while it takes the gradient from the subsequent level and changes its sign before passing it to the preceding layer during the backpropagation. During the forward propagation, the GRL can be ignored. During the backpropagation, the GRL makes the sign of gradient on $h_b^{(4)}$ reverse, which solves the problems caused by the negative sign in Eq. (17).

### B.2 Method Formulation and Implementation

While it is common to use some probability or Bayesian approaches for formulation when one mentions distributions, we actually do not formulate the latent distributions: we are not a generative or parametric method. In fact, the concept of latent distribution is just a notion to help understand our method. Our ultimate goal is to infer which distribution a segment belongs to for best OOD performance. Thus, we do not care what a distribution exactly looks like or even parameterize it since it is not our focus. As long as we can obtain diverse latent distributions, things are all done.

In real implementation, the latent distributions are just represented as domain labels: latent distribution $i$ is also a domain $i$ that certain time series segments belong to, as we stated in the introduction part. Additionally, we also acknowledge that parameterizing the latent distributions may help to get better performance, which can be left for future research.

### B.3 Comparisons to Other Latest Methods

Here, we offer more details on comparisons to other latest methods utilized in the main paper.

- DANN (Ganin et al., 2016) is a method that utilizes the adversarial training to force the discriminator unable to classify domains for better domain-invariant features. It requires domain labels and splits data in advance while ours is a universal method.
- CORAL (Sun & Saenko, 2016) is a method that utilizes the covariance alignment in feature layers for better domain-invariant features. It also requires domain labels and splits data in advance.
- Mixup (Zhang et al., 2018) is a method that utilizes interpolation to generate more data for better generalization. Ours mainly focuses on generalized representation learning.
- GroupDRO (Sagawa et al., 2020) is a method that seeks a global distribution with the worst performance within a range of the raw distribution for better generalization. Ours study the internal distribution shift instead of seeking a global distribution close to the original one.
- RSC (Huang et al., 2020) is a self-challenging training algorithm that forces the network to activate features as much as possible by manipulating gradients. It belongs to gradient operation-based DG while ours is to learn generalized features.
- ANDMask (Parascandolo et al., 2021) is another gradient-based optimization method that belongs to special learning strategies. Ours focuses on representation learning.
- GILE (Qian et al., 2021) is a disentanglement method designed for cross-person human activity recognition. It is based on VAEs and requires domain labels.
- AdaRNN (Du et al., 2021) is a method with a two-stage that is non-differential and it is tailored for RNN. A specific algorithm is designed for splitting. Ours is universal and is differential with better performance.

## C Dataset

### C.1 Datasets Information

Table 4 shows the statistical information on each dataset.

Table 4: Information on datasets.

| Dataset | Subjects | Sensors | Classes | Samples |
|---------|----------|---------|---------|---------|
| EMG | 36 | 1 | 7 | 33,903,472 |
| SPCMD | 2618 | - | 35 | 105,829 |
| WESAD | 15 | 8 | 4 | 63,000,000 |
| DSADS | 8 | 3 | 19 | 1,140,000 |
| USC-HAD | 14 | 2 | 12 | 5,441,000 |
| UCI-HAR | 30 | 2 | 6 | 1,310,000 |
| PAMAP | 9 | 3 | 18 | 3,850,505 |

## C.2 MORE DETAILS ON EMG

Electromyography (EMG) is a typical time-series data that is based on bioelectric signals. We use EMG for gestures Data Set (Lobov et al., 2018) that contains raw EMG data recorded by MYO Thalmic bracelet. The bracelet is equipped with eight sensors equally spaced around the forearm that simultaneously acquire myographic signals. Data of 36 subjects are collected while they performed series of static hand gestures and the number of instances is $40,000 - 50,000$ recordings in each column. It contains 7 classes and we select 6 common classes for our experiments. We randomly divide 36 subjects into four domains (i.e., $0, 1, 2, 3$) without overlapping and each domain contains data of 9 persons.

## C.3 DETAILS ON SENSOR-BASED HUMAN ACTIVITY RECOGNITION DATASET

UCI daily and sports dataset (**DSADS**) (Barshan & Yüksek, 2014) consists of 19 activities collected from 8 subjects wearing body-worn sensors on 5 body parts. USC-SIPI human activity dataset (**USC-HAD**) (Zhang & Sawchuk, 2012) is composed of 14 subjects (7 male, 7 female, aged 21 to 49) executing 12 activities with a sensor tied on the front right hip. **UCI-HAR** (Anguita et al., 2012) is collected by 30 subjects performing 6 daily living activities with a waist-mounted smartphone. **PAMAP** (Reiss & Stricker, 2012) contains data of 18 activities, performed by 9 subjects wearing 3 sensors.

## C.4 DETAILS ON DIFFERENT SETTINGS FOR HUMAN ACTIVITY RECOGNITION

We construct *four* different settings representing different degrees of generalization: (1) **Cross-person generalization**: This setting utilizes DSADS, USC-HAD, PAMAP[10] datasets to construct three benchmarks. Within each dataset, we randomly split the data into four groups and then use three groups as training data to learn a generalized model for the last group. (2) **Cross-position generalization**: this setting uses DSADS dataset and data from each position denotes a different domain. Each sample contains three sensors with nine dimensions. We treat one position as the test domain while the others are for training. (3) **Cross-dataset generalization**: this setting uses all four datasets, and each dataset corresponds to a different domain. Six common classes are selected. Two sensors from each dataset that belong to the same position are selected and data is down-sampled to have the same dimension. (4) **One-Person-To-Another**. This setting adopts DSADS, USC-HAD, and PAMAP datasets. In each dataset, we randomly select four pairs of persons where one is the training and the other is the test.

## C.5 DATA PREPROCESSING

We will introduce how we preprocess data and the final dimension of data for experiments here. We mainly utilize the sliding window technique, a common technique in time-series classification, to split data. As its name suggests, this technique involves taking a subset of data from a given array or sequence. Two main parameters of the sliding window technique are the window size, describing a subset length, and the step size, describing moving forward distance each time.

For EMG, we set the window size 200 and the step size 100, which means there exist $50\%$ overlaps between two adjacent samples. We normalize each sample with $\tilde{\mathbf{x}} = \frac{\mathbf{x} - \min \mathbf{X}}{\max \mathbf{X} - \min \mathbf{X}}$. $\mathbf{X}$ contains all $\mathbf{x}$. The final dimension is $8 \times 1 \times 200$.

---

[10]We do not use UCI-HAR in cross-person setting since its baseline is good enough.

For Speech Commands, we follow (Kidger et al., 2020).

For WESAD, we utilize the same preprocessing as EMG.

Now we give details on all datasets in Cross-person setting. For DSADS, we directly utilize data split by the providers. The final dimension shape is $45 \times 1 \times 125$. $45 = 5 \times 3 \times 3$ where 5 means five positions, the first 3 means three sensors, and the second 3 means each sensor has three axes. For USC-HAD, the window size is 200 and the step size is 100. The final dimension shape is $6 \times 1 \times 200$. For PAMAP, the window size is 200 and the step size is 100. The final dimension shape is $27 \times 1 \times 200$. For UCI-HAR, we directly utilize data split by the providers. The final dimension shape is $6 \times 1 \times 128$.

For Cross-position, we directly utilize samples obtained from DSADS in Cross-person setting. Since each position corresponds to one domain, a sample is split into five samples in the first dimension. And the final dimension shape is $9 \times 1 \times 125$.

For Cross-dataset, we directly utilize samples obtained in Cross-person setting. To make all datasets share the same label space and input space, we select six common classes, including WALKING, WALKING UPSTAIRS, WALKING DOWNSTAIRS, SITTING, STANDING, LAYING. In addition, we down-sample data and select two sensors from each dataset that belong to the same position. The final dimension shape is $6 \times 1 \times 50$.

For One-Person-To-Another, we randomly select four pairs of persons from DSADS, USC-HAD, and PAMAP respectively. Four tasks are $1 \rightarrow 0, 3 \rightarrow 2, 5 \rightarrow 4$, and $7 \rightarrow 6$. Each number corresponds to one subject. And the final dimension shape is $45 \times 1 \times 125, 6 \times 1 \times 200$, and $27 \times 1 \times 200$ for DSADS, USC-HAD, and PAMAP respectively.

As we can see, samples in EMG, WESAD, and HAR all have more than one channel (the first dimension shape), which means they are all multivariate.

## C.6 Details on Domain Splits

We introduce how we split data here.

Since Speech Commands is a regular task, we just randomly split the entire dataset into a training dataset, a validation dataset, and a testing dataset.

We mainly focus on EMG, WESAD, and HAR, and we construct domains for OOD tasks. We denote subjects of a dataset with $0 - s_n$, where $s_n$ is the number of subjects in the dataset. For example, there are 36 subjects in EMG and we utilize $0, 1, 2, \cdots, 35$ to denote data of them respectively.

Table 5 shows the initial domain splits of EMG, WESAD, and all datasets for HAR in Cross-person setting. We just want to make each domain has a similar number of samples in one dataset. As noted in the main paper, we also utilize 0, 1, 2, and 3 to represent different domains but they have different meanings with subjects. When conducting experiments, we take one domain as the testing data and the others as the training data. Our method is not influenced by the splits of the training data since we do not need the domain labels.

Table 5: Initial domain splits.

| Dataset | 0 | 1 | 2 | 3 |
|---------|-----|-----|--------|--------|
| EMG | 0-8 | 9-17 | 18-26 | 27-35 |
| WESAD | 0-3 | 4-7 | 8-11 | 12-14 |
| DSADS | 0,1 | 2,3 | 4,5 | 6,7 |
| USC-HAD | 0,1,2,11 | 3,5,6,9 | 7,8,10,13 | 4,12 |
| PAMAP | 2,3,8 | 1,5 | 0,7 | 4,6 |

## D  Network Architecture and Hyperparameters

For the architecture, the model contains two blocks, and each has one convolution layer, one pooling layer, and one batch normalization layer. A single-fully-connected layer is used as the bottleneck layer while another fully-connected layer serves as the classifier. All methods are implemented with

PyTorch (Paszke et al., 2019). The maximum training epoch is set to 150. The Adam optimizer with weight decay $5 \times 10^{-4}$ is used. The learning rate for GILE is $10^{-4}$. The learning rate for the rest methods is $10^{-2}$ or $10^{-3}$. (For Speech Commands with MatchBoxNet3-1-64, we also try the learning rate, $10^{-4}$.) We tune hyperparameters for each method.

For the pooling layer, we utilize MaxPool2d in PyTorch. The kernel size is $(1, 2)$ ad the stride is 2. For the convolution layer, we utilize Conv2d in PyTorch. Different tasks have different kernel sizes and Table 6 shows the kernel sizes.

Table 6: The kernel size of each benchmark.

| | | | | |
|---|---|---|---|---|
| EMG | (1,9) | | DSADS | (1,9) |
| SPEECHCOMMANDS | (1,9) | Cross-Person | USC-HAD | (1,6) |
| WESAD | (1,9) | | PAMAP2 | (1,9) |
| Cross-position | (1,9) | One-Person-To-Another | The same as Cross-Person | |
| Cross-dataset | (1,6) | | | |

# E  EVALUATION METRICS

We utilize average accuracy on the testing dataset as our evaluation metrics for all benchmarks. Average accuracy is the most common metric for DG and it can be computed as the following,

$$Acc = \frac{\sum_{(\mathbf{x},y)\in\mathcal{D}^{te}} I_y(y*)}{\#|\mathcal{Y}^{te}|},$$
$$y* = \arg\max h(\mathbf{x}).$$

$I_y(y*)$ is an indicator function. If $y = y*$, it equals 1, otherwise it equals 0. $\#|\cdot|$ represents the number of the set. $h$ is the model to learn. Please note that $\mathcal{X}^{te}$ has a different distribution from $\mathcal{X}^{tr}$ for EMG and HAR. And $\mathbf{x}$ has been preprocessed and each sample is a segment.

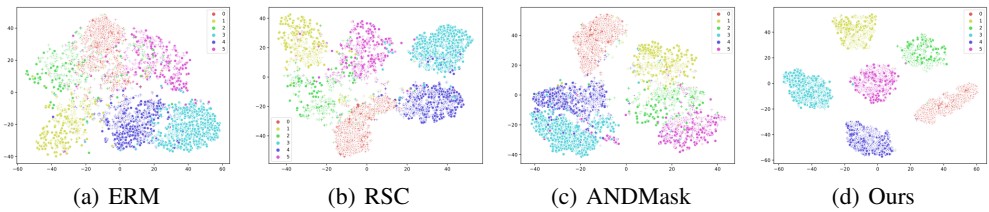

| (a) ERM | (b) RSC | (c) ANDMask | (d) Ours |

Figure 9: Visualization of the t-SNE embeddings for classification on EMG. Different colors correspond to different classes while different shapes correspond to different domains.

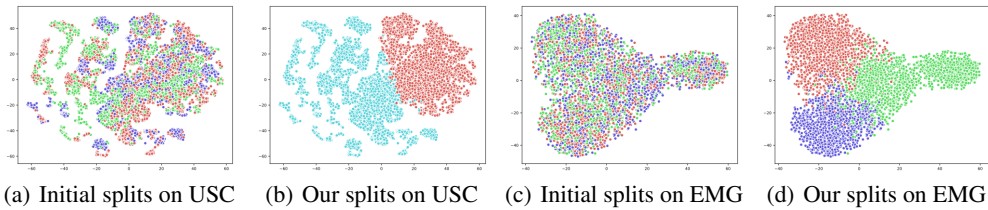

| (a) Initial splits on USC | (b) Our splits on USC | (c) Initial splits on EMG | (d) Our splits on EMG |

Figure 10: Visualization of the t-SNE embeddings for domain splits where different colors represent different domains.

## F MORE EXPERIMENTAL RESULTS

### F.1 VISUALIZATION STUDY

We show more visualization study in this part. As shown in Figure 9(a) and Figure 9(b), both ERM and RSC also cannot obtain fine domain-invariant representations and our method still achieves the best domain-invariant representations. As shown in Figure 10, compared with initial domain splits, latent sub-domains generated by our method are better separated.

### F.2 PARAMETER SENSITIVITY

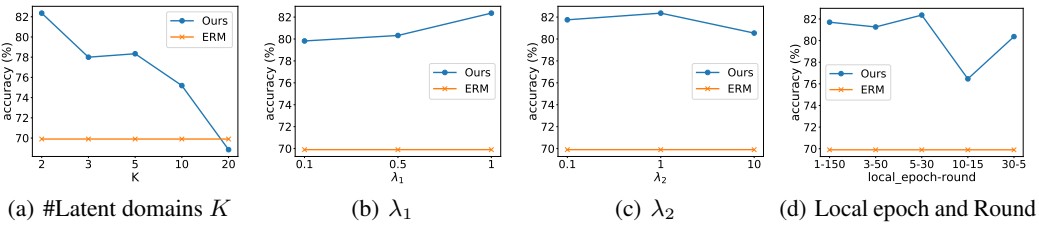

(a) #Latent domains $K$  (b) $\lambda_1$  (c) $\lambda_2$  (d) Local epoch and Round

Figure 11: Parameter sensitivity analysis (EMG).

There are mainly four hyperparameters in our method: $K$ which is the number of latent sub-domains, $\lambda_1$ for the adversarial part in step 3, $\lambda_2$ for the adversarial part in step 4, and local epochs and total rounds. For fairness, the product of local epochs and total rounds is the same value. We evaluate the parameter sensitivity of our method in Figure 11 where we change one parameter and fix the other to record the results. From these results, we can see that our method achieves better performance in a wide range, demonstrating that our method is robust.

### F.3 $\mathcal{H}$-DIVERGENCE

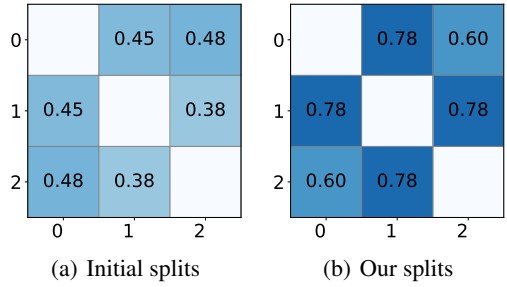

(a) Initial splits   (b) Our splits

Figure 12: $\mathcal{H}$-divergence among domains with initial splits and our splits on EMG.

Figure 12 shows $\mathcal{H}$-divergence among domains with initial splits and our splits on EMG, which demonstrates our splits have larger $\mathcal{H}$-divergence and thereby can bring better generalization.

### F.4 THE INFLUENCE OF ARCHITECTURES

To ensure that our method can work with different sizes of models, we add some more experiments with more complex or simpler architectures. As shown in Table 7, where small, medium, and large indicate the different model sizes (our paper uses the medium), we see a clear picture that model sizes influence the results, and our method also achieves the best performance. Small corresponds to the model with one convolutional layer, Medium corresponds to the model with two convolutional layers, and Large corresponds to the model with four convolutional layers. For most methods, more complex models bring better results.

Table 7: Results on EMG dataset with different model sizes.

| Model Size | Small | | | | | Medium | | | | | Large | | | | |
|---|---|---|---|---|---|---|---|---|---|---|---|---|---|---|---|
| Target | 0 | 1 | 2 | 3 | AVG | 0 | 1 | 2 | 3 | AVG | 0 | 1 | 2 | 3 | AVG |
| ERM | 56.6 | 65.7 | 65.3 | 61.8 | 62.3 | 62.6 | 69.9 | 67.9 | 69.3 | 67.4 | 61.2 | 78.8 | 68.8 | 64.6 | 68.4 |
| DANN | 65.3 | 69.3 | 63.6 | 62.9 | 65.3 | 62.9 | 70.0 | 66.5 | 68.2 | 66.9 | 63.0 | 72.7 | 69.4 | 68.5 | 68.4 |
| CORAL | 66.9 | 74.9 | 70.8 | 73.2 | 71.4 | 66.4 | 74.6 | 71.4 | 74.2 | 71.7 | 67.7 | 77.0 | 72.7 | 71.8 | 72.3 |
| Mixup | 56.8 | 61.0 | 68.1 | 67.2 | 63.2 | 60.7 | 69.9 | 70.5 | 68.2 | 67.3 | 66.3 | 81.1 | 71.2 | 69.6 | 72.0 |
| GroupDRO | 64.9 | 75.0 | 71.6 | 69.1 | 70.1 | 67.6 | 77.5 | 73.7 | 72.5 | 72.8 | 66.3 | 79.3 | 74.9 | 71.3 | 73.0 |
| RSC | 62.7 | 73.2 | 67.6 | 64.0 | 66.9 | 70.1 | 74.6 | 72.4 | 71.9 | 72.2 | 65.1 | 76.8 | 72.2 | 67.7 | 70.4 |
| ANDMask | 62.4 | 66.0 | 66.3 | 65.6 | 65.1 | 66.6 | 69.1 | 71.4 | 68.9 | 69.0 | 65.7 | 78.1 | 72.1 | 71.9 | 71.9 |
| DIVERSIFY | **69.8** | **77.3** | **74.4** | **74.4** | **74.0** | **71.7** | **82.4** | **76.9** | **77.3** | **77.1** | **72.0** | **86.6** | **78.5** | **78.9** | **79.0** |

Table 8: Results on EMG dataset with Transformer.

| Target | 0 | 1 | 2 | 3 | AVG |
|---|---|---|---|---|---|
| ERM | 71.7 | 83.8 | 76.1 | 78.0 | 77.4 |
| DANN | 72.7 | 82.3 | 77.6 | 77.3 | 77.5 |
| GroupDRO | 72.1 | 84.4 | 77.5 | 78.3 | 78.1 |
| RSC | 72.1 | 83.7 | 77.6 | 78.2 | 77.9 |
| Ours | **75.6** | **86.2** | **79.4** | **79.7** | **80.2** |

We also try Transformer (Vaswani et al., 2017) as the backbone for comparisons. As shown in (Zhang et al., 2022a), Transformer often has a better generalization ability compared to CNN, which implies improving with Transformer is more difficult. From Table 8, we can see that each method with Transformer has a remarkable improvement on EMG. Compared to ERM, DANN almost has no improvement but ours still has further improvements and achieves the best performance. To further validate the advantage of our method, we perform the experiments on a more difficult task, i.e. the first task of cross-dataset where distribution gaps are larger. As shown in Figure F.4, our method still achieves the best performance in this more difficult situation while DANN even performs worse than ERM, which demonstrates the importance of more accurate sub-domain labels.

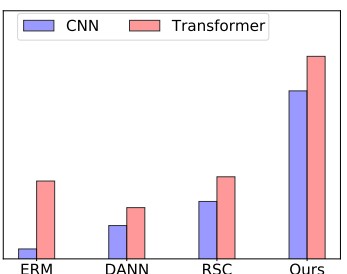

Figure 13: Results on the first task in Cross-dataset generalization using Transformers.

Overall, for all architectures, our method achieves the best performance.

Table 9: Time costs of different methods (s).

| method | ERM | DANN | CORAL | Mixup | GroupDRO | RSC | ANDMask | Ours |
|---|---|---|---|---|---|---|---|---|
| time | 305.98 | 328.62 | 357.54 | 2042.93 | 340.03 | 321.51 | 379.87 | 356.8 |

## F.5 Time Complexity and Convergence Analysis

We also provide some analysis on time complexity and convergence. Since we only optimize the feature extractor in Step 2, our method does not cost too much time. And the results in Table 9 prove this argument empirically.

The convergence results are shown in Figure F.5. Our method is convergent. Although there are some little fluctuations, these fluctuations exist widely in all domain generalization methods due to different distributions of different samples.

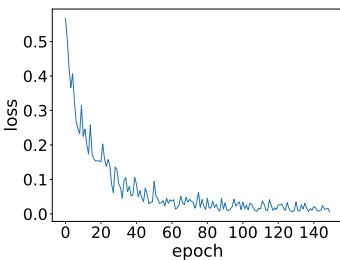

Figure 14: Convergence results on EMG.

