# OpenReview forum: "Out-of-distribution Representation Learning for Time Series Classification"
_ICLR.cc/2023/Conference — ICLR 2023 poster_

### Official Review · Reviewer_MfQf · 2022-10-17

**Confidence:** 2
**Correctness:** 3
**Technical Novelty And Significance:** 2
**Empirical Novelty And Significance:** 3
**Recommendation:** 5

**Clarity, Quality, Novelty And Reproducibility:**

The math is not completely clear to me, I don't fully understand the method or loss function proposed.

**Strength And Weaknesses:**

Strength
Improved performance

Weakness
Unclear how the model works
Lack of visualizations
Motivation is not completely correct
Unclear if improved performance is state-of-the-art

**Summary Of The Paper:**

The paper proposes a new machine learning model, and training objective, for classifying signal data from speech, wearables, EMG, and sensors. I don't fully understand the proposed model (see questions for clarification below). But it outperforms previous methods in the field.

"While most efforts have predominantly resided in image classification, OOD in time series remains underexplored and more challenging." I don't think it's underexplored, there's entire research fields dedicated to sequential extrapolation. Look up DYCK languages or most of the papers published by Michael Hahn in 2019-2020. Moreover, if the OOD is in reference to signal processing, I am sure the field of reinforcement learning and robotics are dealing with similar challenges of OOD.

After reading the introduction, I still don't understand what "DIVERSIFY" does, other than it's adversarial and models the underlying distribution.

"Surprisingly, we even find temporal distribution shifts in our experiments (Figure 6) that distributions of one person can also change at various times" Do you collect your own dataset? What kind of experiments are you referring to?

"a time series may consist of K2 unknown latent domains3 rather than a fixed one" The way the citation works make it seems like it says K squared.

What are Hc, Hb, and Hf - do you have more detailed explanations of them somewhere? Also, please give a more elaborate explanation of (2).

"At the first iteration, there is no domain label d′ and we simply initialize d′ = 0 for all samples. We treat per category per domain as a new class with label s ∈ {1, 2, · · · , S}. We have S = K × C where K is the pre-defined number of latent distributions that can be tuned in experiments. We perform pseudo domain-class label assignment to get discrete values for supervision: s = d′ × C + y."
I don't understand this. Do you choose a set of sub-sequences by hand, and then force your model to iteratively use more finegrained subsequences?

"Latent Distribution Characterization" Is this some type of hard attention?

Figure 7 is not covered enough, I don't have a strong intuition for why your model works. I would like to see specific examples of data where your model performs better.



**Summary Of The Review:**

I am not familiar with this field and the paper does not make it easier to understand. The authors argue that their focus on OOD on sequences is novel, however, I don't see how OOD signal processing is a new field of research. I need a better motivation. It is not clear to me from the text if the models they are testing against are state-of-the-art, or if they are reimplementations.

---

> ### Author Response · Authors · 2022-11-11
> **Response to Reviewer MfQf (1)**
>
> Thanks for your acknowledgment in our *thorough evaluation*! We see that your main concerns are on more explanations. Now we answer them here.
>
> 1.  "While most efforts have predominantly resided in image classification, OOD in time series remains underexplored and more challenging." I don't think it's underexplored, there's entire research fields dedicated to sequential extrapolation.
>
> The word 'under explored' might be inaccurate since there exists some work in the field, e.g. [1] (2021), [2] (2022) listed by reviewer SWyN, AdaRNN (2021), and GILE (2021). We will replace 'under explored' with 'less studied' in the later version. Please note that ours achieves the best performance in this challenging setting. By the way, there exist some differences between this research field and reinforcement learning. The former field pays more attention to some specific tasks, e.g. classification of walking or running.
>
> 2. After reading the introduction, I still don't understand what "DIVERSIFY" does, other than it's adversarial and models the underlying distribution.
>
> 'DIVERSIFY' is just a *name* to illustrate some characteristics (diversify and distribution matching) of our method. We hope that our method can obtain more diversified sub-domains and thereby learn more generalized features. Theoretical insights and empirical experiments demonstrate that we get it. It is just a name. We can change it if you think it is not feasible:)
>
> 3. "Surprisingly, we even find temporal distribution shifts in our experiments (Figure 6) that distributions of one person can also change at various times" Do you collect your own dataset? What kind of experiments are you referring to?
>
> No, we do not collect our own datasets. These findings are directly from the experiments: We utilize public benchmarks and construct OOD settings by ourselves. In Fig 6(a), we just want to illustrate that there is more than one latent distribution for one person's walking activities. Fig 1 and Fig 6(b) describe the similar phenomenon in different data samples.
>
> 4. "a time series may consist of K2 unknown latent domains3 rather than a fixed one" The way the citation works make it seems like it says K squared.
>
> We are sorry for this confusing expression. '2' here means a *footnote* in our paper. We will slightly change the location of 2 in the later version.
>
> 5. What are Hc, Hb, and Hf - do you have more detailed explanations of them somewhere? Also, please give a more elaborate explanation of (2).
>
> Hc, Hb, and Hf correspond to the classification net, the bottleneck net, and the feature net respectively. They are all some part of the whole net. In Eq (2), we want to compute the centroid of each cluster with predictions from the net, which is similar to K-Means. The denominator is for normalization while the numerator is to compute centers.
>
> 6. "At the first iteration, there is no domain label d′ and we simply initialize d′ = 0 for all samples. We treat per category per domain as a new class with label s ∈ {1, 2, · · · , S}. We have S = K × C where K is the pre-defined number of latent distributions that can be tuned in experiments. We perform pseudo domain-class label assignment to get discrete values for supervision: s = d′ × C + y." I don't understand this. Do you choose a set of sub-sequences by hand, and then force your model to iteratively use more finegrained subsequences?
>
> No, we do not choose a set of sub-sequences by hand. We just offer initiation and then the model performs all things automatically.

---

> > ### Comment · Reviewer_MfQf · 2022-11-17
> > **Feedback**
> >
> > General:
> > They authors didn't address much of what I asked. My rating remains the same. The authors should consider having a native english speaker help with reviewing their paper writing. It could be more concise.
> >
> > See specifics below:
> >
> > "While most efforts have predominantly resided in image classification, OOD in time series remains
> > less studied and more challenging. " This doesn't sit well with me, computer vision is a big field within the machine learning community, but time series extrapolation is not less studied, it is a very studied field. If you want to highlight that some research is missing in your subdomain, then you need to be more specific than just stating "time-series". Instead, you need to define what specific type of time-series you're looking at, what makes the challenging, and how those specific challenges represents OOD that can be hard to solve with current technology.
> >
> >
> > "'DIVERSIFY' is just a name to illustrate some characteristics (diversify and distribution matching) of our method. We hope that our method can obtain more diversified sub-domains and thereby learn more generalized features. Theoretical insights and empirical experiments demonstrate that we get it. It is just a name. We can change it if you think it is not feasible:)"
> >
> > That's not what I'm saying, DIVERSIFY is probably a fine name. I'm saying that you don't describe your solution well enough for me to get a good grasp of it from the introduction. You're not concise enough in your introduction.
> >
> > "No, we do not collect our own datasets. These findings are directly from the experiments: We utilize public benchmarks and construct OOD settings by ourselves." This is confusing to me, I still don't understand what you do from a dataset perspective
> >
> > "Hc, Hb, and Hf correspond to the classification net, the bottleneck net, and the feature net respectively. They are all some part of the whole net. In Eq (2), we want to compute the centroid of each cluster with predictions from the net, which is similar to K-Means. The denominator is for normalization while the numerator is to compute centers."
> >
> > I want to see the math clearly describing what Hc, Hb, and Hf are in the paper.
> >
> > "No, we do not choose a set of sub-sequences by hand. We just offer initiation and then the model performs all things automatically."
> > If I don't understand this, others might not understand it either. You should reconsider how you formulate your explanations.
> >
> > "No, it is not some type of hard attention. We just want to split data into different groups. Data samples in the same group share the similar distributions. We have released the code in supplementary. Please refer to it."
> > Same as above, if I don't understand it I need a better explanation. Unless it's a common practice in the field it needs an introduction.
> >
> > "Figure 7(a) and 7(b) demonstrate that our domain splits are better than the initial artificial splits, which is the reason why ours performs better. Figure 7(c) and 7(d) demonstrate that our method can learn more generalized features which are more related to classification and are not influenced by different distributions. "
> > It smells like cherry picking to me.
> >
> > "Unclear how the model works"
> > Same as above, if it's unclear to me it might also be unclear to others. You should update your paper.

---

> > > ### Author Response · Authors · 2022-11-18
> > > **More clarifications and explanations (1/2)**
> > >
> > > Dear reviewer MfQf,
> > >
> > > We are sorry for not addressing your concerns in previous response. We now give more explanations and clarifications to help you better understand our contributions and what we do.
> > >
> > > 1. > The authors should consider having a native english speaker help with reviewing their paper writing. It could be more concise.
> > >
> > > We agree that it is never enough to polish a paper. While it is relieved to see that *reviewer SWyN* said **'the paper is clearly written, with a detailed supplementary material.'** and *reviewer oHGw* said **'The work is clearly written and easy to follow.'**, we think there could be some misunderstandings in your reading, which makes it hard to understand this paper. We will continuously work hard to polish it.
> > >
> > >
> > > 2. > This doesn't sit well with me, computer vision is a big field within the machine learning community, but time series extrapolation is not less studied, it is a very studied field. If you want to highlight that some research is missing in your subdomain, then you need to be more specific than just stating "time-series". Instead, you need to define what specific type of time-series you're looking at, what makes the challenges, and how those specific challenges represent OOD that can be hard to solve with current technology.
> > >
> > > This is totally a **misunderstanding**:
> > > - We **never** say 'time series extrapolation is not less studied'. Instead, we say '**OOD** in time series...'. OOD does not equal to extrapolation.
> > > - We are dealing with general time series classification problems, as recognized by reviewer oHGw and SWyN. This is because OOD in time series are not well studied, as suggested by SWyN and he/she even suggested severall related work.
> > > * We have listed some examples in the main paper: 'Example applications include sensor-based human activity recognition, Parkinson’s disease diagnosis, and electronic power consumption (Fawaz et al., 2019). One important nature of time series is the non-stationary property, indicating that its statistical features are changing over time.'
> > > * Our contributions, which are also recognized by other reviewers: 'We propose to model time series from the distribution perspective to handle its dynamically changing distributions; more precisely, to learn out-of-distribution (OOD) representations for time series that can generalize to unseen distributions.'
> > >
> > > We also offer a specific example in Figure 1. For a more basic introduction on OOD, please refer to [1].
> > >
> > > [1] Jindong Wang, Cuiling Lan, Chang Liu, Yidong Ouyang, Wenjun Zeng, and Tao Qin. Generalizing to unseen domains: A survey on domain generalization. IEEE Transactions on Knowledge and Data Engineering (TKDE), 2022b.
> > >
> > > 3. > That's not what I'm saying, DIVERSIFY is a fine name. I'm saying that you don't describe your solution well enough for me to get a good grasp of it from the introduction. You're not concise enough in your introduction.
> > >
> > > If you are fine with the name DIVERSIFY, then the name is not important anymore. The most important part is *how we do it*, which is explained in *Figure 2* in the main paper: first, pre-process time series; second, learn the worst-case distribution scenario where all distributions are diverse (i.e., "DIVERSIFY"); third, learn the domain-invariant features from those diverse distributions; forth, update features and go back to step 2.
> > >
> > > 4. > This is confusing to me, I still don't understand what you do from a dataset perspective.
> > >
> > > We never perform data collection on our own, but just preprocessing public datasets. For specific operations, please refer to Experiments and Appendix. We also offer the **code** in supplementary which we think can better help you understand what we do.
> > >
> > > 5. > I want to see the math clearly describing what Hc, Hb, and Hf are in the paper.
> > >
> > > Hc, Hb, and Hf represent various parts of networks, i.e., they are basically **functions**, implemented by neural networks, as written in the main paper. For more mathematical expressions, please refer to some DL books, e.g. [2].
> > >
> > > [2] Goodfellow, Ian, Yoshua Bengio, and Aaron Courville. Deep learning. MIT press, 2016.
> > >
> > > 6. > If I don't understand this, others might not understand it either. You should reconsider how you formulate your explanations.
> > >
> > > From our side, this step is a common training process of deep learning, we only offer initial values and then we *do not* intervene the training process. We also offer the **code** in supplementary which we think can better help you understand what we do. This is also well recognized by other reviewers such as oHGw and SWyN.
> > >
> > > 7. > Same as above, if I don't understand it I need a better explanation. Unless it's a common practice in the field it needs an introduction.
> > >
> > > OOD is a common and challenging setting than traditional non-OOD ones. The survey work [1] offers more on OOD. Technically speaking, OOD and attention focus on different fields: one is general ML problem setting, the other is used in transformers and is a basic structure.

---

> > > ### Author Response · Authors · 2022-11-18
> > > **More clarifications and explanations (2/2)**
> > >
> > > 8. > "Figure 7(a) and 7(b) demonstrate that our domain splits are better than the initial artificial splits, which is the reason why ours performs better. Figure 7(c) and 7(d) demonstrate that our method can learn more generalized features which are more related to classification and are not influenced by different distributions. " It smells like cherry picking to me.
> > >
> > > Please refer to [1] for more descriptions on definitions, methods, and problems in OOD. Domain-invariant features are basic knowledge in OOD generalization which you should know about.
> > >
> > > 9. > "Unclear how the model works" Same as above, if it's unclear to me it might also be unclear to others. You should update your paper.
> > >
> > > Apparently, there are some basic knowledge about deep learning and OOD generalization that got you confused. While favored by other reviews, we think you should pay more attention to INTRODUCTION and THEORETICAL INSIGHTS for motivations, METHODOLOGY for how we operate, and EXPERIMENTS for how it works.
> > >
> > > In summary, we are happy to explain them to you if you have any other questions. Thanks for your time!

---

> ### Author Response · Authors · 2022-11-11
> **Response to Reviewer MfQf (2)**
>
>
> 7. "Latent Distribution Characterization" Is this some type of hard attention?
>
> No, it is *not* some type of hard attention. We just want to split data into different groups. Data samples in the same group share the similar distributions. We have released the code in supplementary. Please refer to it.
>
> 8. Figure 7 is not covered enough, I don't have a strong intuition for why your model works. I would like to see specific examples of data where your model performs better. Lack of visualizations
>
> Figure 7(a) and 7(b) demonstrate that our domain splits are better than the initial artificial splits, which is the reason why ours performs better. Figure 7(c) and 7(d) demonstrate that our method can learn more generalized features which are more related to classification and are not influenced by different distributions. We also provide more visualization studies in Appendix (Figure 9 and Figure 10). These examples all demonstrate that our method can obtain more accurate sub-domains and thereby better class-related features.
>
> 9. Unclear how the model works
>
> We update the feature extractor using the proposed pseudo domain-class labels as the supervision for fine-grained features. Then according to the feature net, we identify the domain label for each instance to acquire the latent distribution information. We utilize class label information to learn class-invariant features and thereby obtain pseudo domain labels. And we try to make sub-domains diverse. Finally, we utilize pseudo domain labels to learn domain-invariant representations in an adversarial way and train a generalizable model. These three steps process several rounds similar to the EM algorithm.
>
> We released the code for reproducibility.
>
> 10. Motivation is not completely correct
>
> - One important nature of time series is the non-stationary property, indicating that its statistical features are changing over time. The model trained with data from distribution A often performs terribly on data from distribution B. If no attention is paid to exploring latent distributions inside the data (i.e., sub-domains), predictions may fail in face of diverse sub-domain distributions. Theoretical results provide motivational insights from another point.
> - As another evidence to show the importance of OOD on time series, *Reviewer SWyN* also provides two papers [1] and [2] which can prove the importance. And we have compared these works in our revised paper.
>
> 11. Unclear if improved performance is state-of-the-art
>
> Since this field is still less studied, fewer results can be directly utilized. Therefore, we reimplement some state-of-the-art methods, e.g. ANDMask, AdaRNN, and GILE. Moreover, we add some comparisons to DFDG and CCDG mentioned by Reviewer SWyN (please refer to the feedback to their review).
>
> 12. It is not clear to me from the text if the models they are testing against are state-of-the-art, or if they are reimplementations.
>
> Yes, we reimplement these latest methods in our experiments since there exist few results. We also included more comparison methods as suggested by reviewer SWyN and using Transformers. The comparison is for SOTA.
>
> We hope your concerns will be resolved and the rating of the paper can be increased accordingly.
> If there still exist some other problems, please tell us and we are glad to offer more explanations. Thank you!

---

### Official Review · Reviewer_EJKA · 2022-10-25

**Confidence:** 4
**Correctness:** 3
**Technical Novelty And Significance:** 3
**Empirical Novelty And Significance:** 3
**Recommendation:** 8

**Clarity, Quality, Novelty And Reproducibility:**

The work is of high quality. However, there's no provided code for reproducibility.

**Strength And Weaknesses:**

S
+ Strong results in an array of diverse timeseries tasks
+ Extensive experimentation with different settings across people/positions/datasets
+ Intuitive explanation of why the components of the method should be there (along with ablations)

W
- Lack of comparisons with other backbones like Transformers

**Summary Of The Paper:**

This paper presents a representation learning model for OOD timeseries data. By building on top of DANN, it applies clustering in order to identify domains along with pseudo domain-class labels and adversarial self-supervised pseudo labeling to obtain the pseudo domain labels. Extensive experiments are conducted on various datasets with different OOD settings showing that the proposed framework outperforms two closely related baselines and other baselines from the domain generalization literature.

**Summary Of The Review:**

This is a very well designed paper with strong results. My only concerns are about exploring different backbone architectures such as Transformers and lack of code.

---

> ### Author Response · Authors · 2022-11-11
> **Response to Reviewer EJKA**
>
> Thanks for your acknowledgment on our paper! We see that your main concerns are on experiments comparison and code open source. Now we answer them here.
>
> 1. Lack of comparisons with other backbones like Transformers.
>
> - We mainly adopt this simple feature net according to [1]. In the I.I.D setting, this feature net has satisfactory results. We also utilize some other feature nets in our paper, e.g. MatchBoexNet for Speech Commands (Fig 3) and larger feature nets for EMG (Tab 7).
>     - [1] Wang, Jindong, et al. "Deep learning for sensor-based activity recognition: A survey." Pattern recognition letters 119 (2019): 3-11.
> - To further alleviate your concerns, we utilize the **transformer** as the feature net for more comparisons. From the table below, we can see that ours still achieves the best performance. For more analysis, please refer to Sec F.4 in the supplementary.
>
> |                 |  EMG  |       |          |       |       |
> |-----------------|:-----:|:-----:|:--------:|:-----:|:-----:|
> | Target                | 0     | 1     | 2        | 3     | AVG   |
> | ERM             | 71.7  | 83.8  | 76.1     | 78.0  | 77.4  |
> | DANN            | 72.7  | 82.3  | 77.6     | 77.3  | 77.5  |
> | GroupDRO        | 72.1  | 84.4  | 77.5     | 78.3  | 78.1  |
> | RSC             | 72.1  | 83.7  | 77.6     | 78.2  | 77.9  |
> | Ours            | 75.6  | 86.2  | 79.4     | 79.7  | 80.2  |
> | Cross-dataset   | ERM   | DANN  | GroupDRO | RSC   | Ours  |
> |  0 | 35.98 | 32.23 | 30.65    | 36.58 | 53.57 |
>
> 2.  Reproducibility.
>
> To eliminate your concerns, we will upload our codes as the part of supplementary in the later version.
>
> We hope your concerns will be resolved and the rating of the paper can be increased accordingly. Thank you!

---

### Official Review · Reviewer_oHGw · 2022-10-26

**Confidence:** 3
**Correctness:** 3
**Technical Novelty And Significance:** 3
**Empirical Novelty And Significance:** 3
**Recommendation:** 5

**Clarity, Quality, Novelty And Reproducibility:**

The clarity and quality is good, while originality is limited. Reproducibility may be hard as no code is given.

**Strength And Weaknesses:**

* Strength

(i) The work is clearly written and easy to follow.

(ii) The design is sensible, as it is widely seen from other OOD work beyond time-series data.

(iii) Sec 2.3 is insightful, esp. linking Eq. (7) to the step 3 of the proposed algorithm.

* Weakness

(i) The novelty is quite limited, as the techniques used here are largely adapted from existing work.

(ii) For the experiment, it is not clear why such a simple feature net (two-conv) is used, given that stronger backbones exist.

(iii) Why there is an inconsistency over methods for different datasets? i.e., some methods are not evaluated for certain datasets/tasks.

**Summary Of The Paper:**

In this work, the authors propose DIVERSIFY, an OOD representation learning algorithm for time series classification. It has two iterative steps: (i) learn to segment the time series data into several latent sub-domains by maximizing the distribution gap and (ii)  learn domain-invariant representations by reducing the distribution divergence between the obtained latent domains. The experiments on several time series classification tasks showed that the proposed method outperforms other baselines.

**Summary Of The Review:**

Overall I think it is a good work, though there are some flaws in experiment section should be addressed.

---

> ### Author Response · Authors · 2022-11-11
> **Response to Reviewer oHGw**
>
> Thanks for your acknowledgment in our *writing, design, and theoretical insight*! We see that your main concerns are on details of some claims and experiments. Now we answer them here.
>
> 1. The novelty is quite limited, as the techniques used here are largely adapted from existing work.
>
> Please note that although some modules are popular and common, we are **NOT** simple applications of them: we propose our novel **modifications** according to our problem. We propose a method to learn generalized representation for time series classification without domain labels. Our main contributions are exploiting sub-domains, learning generalized representation, and finally obtaining a model that can perform well on out-of-distribution data. We do not simply adapt our method from existing work, but design each module for specific goals.  We propose pseudo domain-class, adapted class-invariant feature learning, and adapted domain-invariant feature learning  for fine-grained feature, pseudo label generation, and classification respectively. This framework is novel and clear. Theoretical insight offers the rationality of our method from another perspective.
>
> 2. For the experiment, it is not clear why such a simple feature net (two-conv) is used, given that stronger backbones exist.
>
> - We mainly adopt this simple feature net according to [1]. In the I.I.D setting, this feature net has satisfactory results. We also utilize some stronger feature nets in our paper, e.g. MatchBoexNet for Speech Commands (Fig 3) and larger feature nets for EMG (Tab 7).
>     - [1] Wang, Jindong, et al. "Deep learning for sensor-based activity recognition: A survey." Pattern recognition letters 119 (2019): 3-11.
>
> - To further alleviate your concerns, we utilize the **transformer** as the feature net for more comparisons. From the table below, we can see that ours still achieves the best performance. For more analysis, please refer to Sec F.4 in the supplementary.
>
> |                 |  EMG  |       |          |       |       |
> |-----------------|:-----:|:-----:|:--------:|:-----:|:-----:|
> | Target                | 0     | 1     | 2        | 3     | AVG   |
> | ERM             | 71.7  | 83.8  | 76.1     | 78.0  | 77.4  |
> | DANN            | 72.7  | 82.3  | 77.6     | 77.3  | 77.5  |
> | GroupDRO        | 72.1  | 84.4  | 77.5     | 78.3  | 78.1  |
> | RSC             | 72.1  | 83.7  | 77.6     | 78.2  | 77.9  |
> | Ours            | 75.6  | 86.2  | 79.4     | 79.7  | 80.2  |
> | Cross-dataset   | ERM   | DANN  | GroupDRO | RSC   | Ours  |
> |  0 | 35.98 | 32.23 | 30.65    | 36.58 | 53.57 |
>
> 3. Why there is an inconsistency over methods for different datasets? i.e., some methods are not evaluated for certain datasets/tasks.
>
> Some methods are designed specifically for some settings. For example, GILE is proposed for activity recognition generalization across persons. Some methods deeply rely on domain labels, e.g. DANN and CORAL. We try our best to compare our methods with the latest methods.
>
> 4. The clarity and quality is good, while originality is limited. Reproducibility may be hard as no code is given.
>
> To eliminate your concerns, we will upload our codes as the part of supplementary in the later version.
>
> We hope your concerns will be resolved and the rating of the paper can be increased accordingly. Thank you!

---

### Official Review · Reviewer_x6zY · 2022-10-28

**Confidence:** 3
**Correctness:** 3
**Technical Novelty And Significance:** 1
**Empirical Novelty And Significance:** 2
**Recommendation:** 5

**Clarity, Quality, Novelty And Reproducibility:**

the domain invariant feature is quite common in domain adaptation literature beyond DANN, e.g., DAF https://arxiv.org/abs/2102.06828,  and thus be suspicious on the novelty of the method. In addition, worst-case scenarios is considerd in ADA-RNN method and suspicious of the orignality of the concept.

**Strength And Weaknesses:**

## Strength
- consider worst-case scenario and develop the algorithm and modeling end to end.
- extensive real-data experiments to demonstrate the superiority of the method.


## Weakness
- lack of motivation and weak examiniation on the method
 - why OOD is important in time series?
 - how was OOD defined in the time series formally? at least reference? these definition can be different from static data
 - What if all real dataset do not fall in OOD categories the authors defined? any synthetic experiments to support the necessity of the method?
 - the theoretical results is adopted from existing one, why is this specially applicable to time series setting beyond static setting? Without it, it is just OOD method in image classification and need more thoroughly to be compared with those literatures?
 - similary, why existing methods in static classification are not applicable to time series setting?

**Summary Of The Paper:**

This paper propose a method for representation learning robust to out of distribution cases for the times series classification cases.

**Summary Of The Review:**

Even with the propose method end2end and extensive real data experimetns, the contribution is marginal, not strong connection of OOD motivation and supporting evidences. Many things the author claimed 'novel', like perspetive and methodology is variant of existing works and thus marginal, or if not, authors should more thoroughly examine and point out the differences.

---

> ### Author Response · Authors · 2022-11-11
> **Response to Reviewer x6zY**
>
> Thanks for your acknowledgment in our *thorough evaluation*! We see that your main concerns are on details of some claims and innovations. Now we answer them here.
>
> 1. motivation and weak examiniation on the method. why OOD is important in time series?
>
> We have mentioned it in the introduction and we briefly describe it again here.
> - One important nature of time series is the non-stationary property, indicating that its statistical features are changing over time. The model trained with data from distribution A often performs terribly on data from distribution B. If no attention is paid to exploring latent distributions inside the data (i.e., sub-domains), predictions may fail in face of diverse sub-domain distributions.
> - As another evidence to show the importance of OOD on time series, *Reviewer SWyN* also provides two papers [1] and [2] which can prove the importance. And we have compared these works in our revised paper.
>
> [1] Zhang, Wenyu, Mohamed Ragab, and Ramon Sagarna. "Robust domain-free domain generalization with class-aware alignment." ICASSP 2021-2021 IEEE International Conference on Acoustics, Speech and Signal Processing (ICASSP). IEEE, 2021.
>
> [2] Ragab, Mohamed, et al. "Conditional Contrastive Domain Generalization for Fault Diagnosis." IEEE Transactions on Instrumentation and Measurement 71 (2022): 1-12.
>
> 2. how was OOD defined in the time series formally? at least reference? these definition can be different from static data
>
> There might exist some *misunderstandings*. We do NOT emphasize the differences between OOD in time series and static data. 'Static' is not the key point. We only want to illustrate the fact that the distribution of time-series data keeps changing and we have no idea about which domain they belong to. For example, a man walking from A to B can generate data from different distributions due to his changing walking styles.
>
> 3. What if all real dataset do not fall in OOD categories the authors defined? any synthetic experiments to support the necessity of the method?
>
> This is a good question. In fact, OOD is a more general setting than the non-OOD ones since the real world is full of uncertainties: any variation can cause the change of the distribution. Thus, our method is general. As an empirical evidence, our experiments on **Speech Commands dataset** in Sec. 3.2 provides strong support for this: note that Speech Commands is a general dataset that never claimed OOD. But our method can achieve even better performance than SOTA (matchboxnet):)
>
> 4. the theoretical results is adopted from existing one, why is this specially applicable to time series setting beyond static setting? Without it, it is just OOD method in image classification and need more thoroughly to be compared with those literatures?
>
> Yes, you're right: the theoretical result is also applicable to the static setting. One can view the dynamic setting as a more complex version of the static setting, where the basic ingredients do not change. Additionally, please note that the theoretical result is only insight and motivation: It explains why we design the method in this way and what our method can bring. We only want to explain the superiority and rationality of our method from both empirical experiments and theoretical insights.
>
> 5. similary, why existing methods in static classification are not applicable to time series setting?
>
> In fact, some existing methods are also applicable to the time series setting if given domain labels. We compare our method to these methods, e.g. DANN and CORAL. Since artificial domain labels can be inaccurate, ours performs better. Moreover, we add some comparisons to DFDG and CCDG mentioned by Reviewer SWyN.
>
> 6. Novelty.
>
> - Adversarial training is a common and popular technique in the deep learning community. Note that, our method is *NOT* a simple application of DANN, but we **adapt** it for our purposes: We propose pseudo domain-class, adapted class-invariant feature learning, and adapted domain-invariant feature learning  for fine-grained feature, pseudo label generation, and classification respectively. We prove the superiority and rationality of our framework from both empirical experiments and theoretical insights.
> - AdaRNN obtains domain labels in a greedy way which is not end-to-end and can generate inaccurate predictions.  In addition, AdaRNN is tailored for RNN. Experimental results have demonstrated that ours is better.
>
> We hope your concerns will be resolved and the rating of the paper can be increased accordingly. Thank you!

---

### Official Review · Reviewer_SWyN · 2022-11-02

**Confidence:** 4
**Correctness:** 3
**Technical Novelty And Significance:** 3
**Empirical Novelty And Significance:** 3
**Recommendation:** 6

**Clarity, Quality, Novelty And Reproducibility:**

Clarity: the paper is clearly written, with a detailed supplementary material.

Quality: the paper is technically sounds, though some claims are not well supported.

Novelty: The assumption of sub-domains and the ability to detect and adapt these sub-
domains is somewhat novel. However, the ideas of using pseudo domain labels and the
assumption of no domain label existing are existing in the literature

Reproducibility: The paper is reproducible

**Strength And Weaknesses:**

Strength:

The problem of domain generalization for time series classification is significant
and important

A new perspective for the inherit sub-distributions inside the time series data is
insightful

Theorical insights are also provided.
Weaknesses:

The technicality of the idea, while assuming there exists sub-distributions
within each well-defined domain is plausible, it is not clear how the method
differentiates between the class-distributions within each domain (which is also
can be detected as sub-domains) and the original sub-distributions.

Overclaims and Novelty:

The authors claimed that domain generalization for time series
classification is that “under explored” without mentioning in time series
domain generalization methods. However, the exist some domain
generalization methods for time series data such as [1,2]

The authors claim novelty for working and using pseudo domain labels.
However, some domain generalization methods exist that use peudo
domain labels such as [3].

The authors claim “Finally, our work is not a direct application of DG due
to the non-existence of domain labels ” to distinguish their methods from
domain generalization approach. Nevertheless, there exist some
approaches that also doesn’t require domain labels [1, 2, 3]

Experiments:

One key motivation for the paper is the detection of sub-distributions
within each domain. However, all the tested domain generalization
scenarios are clearly usually mapping across clearly separated domains
such as different humans where each human is one domain. Thus, I
cannot see any experiment that support their claim of the existence of
sub-domains.

Missing Time series domain generalization baselines [1,2].

[1] Zhang, Wenyu, Mohamed Ragab, and Ramon Sagarna. &quot;Robust domain-free domain generalization
with class-aware alignment.&quot; ICASSP 2021-2021 IEEE International Conference on Acoustics, Speech
and Signal Processing (ICASSP). IEEE, 2021.

[2] Ragab, Mohamed, et al. &quot;Conditional Contrastive Domain Generalization for Fault Diagnosis.&quot; IEEE
Transactions on Instrumentation and Measurement 71 (2022): 1-12.

[3] Matsuura, Toshihiko, and Tatsuya Harada. &quot;Domain generalization using a mixture of multiple latent
domains.&quot; Proceedings of the AAAI Conference on Artificial Intelligence. Vol. 34. No. 07. 2020.

**Summary Of The Paper:**

This paper proposes a domain generalization method for time series classification to
improve the generalization on new unseen target domain without using domain labels.
Moreover, an identification approach of the sub-distributions inside the data is
proposed. Last, a min-max adversarial approach is presented to find domain invariant
features among the predicted domains and sub-domains. Experimental results on four
different time series datasets show the superiority of the proposed approach.

**Summary Of The Review:**

The paper addresses a challenging yet an important problem of distribution shift for time
series data . The main weaknesses are the incomplete support of some claims, while
missing some important baselines in the related works and experiments.

---

> ### Author Response · Authors · 2022-11-11
> **Response to Reviewer SWyN**
>
> Thanks for your acknowledgment in our *novelty, technical soundness, theoretical insight, and thorough evaluation*! We see that your main concerns are on details of some *claims* and *more baselines and comparisons*. Now we answer them here.
>
> 1. How the method differentiates between the class-distributions within each domain (which can also be detected as sub-domains) and the original sub-distributions.
>
> In a real OOD scenario, domain labels *do not exist*. However, most existing methods require domain labels. And we have to label domains artificially according to some factors, e.g. person. Sometimes these manual labels are inaccurate. From Figure 6(a) and 6(b), we can see that data samples from the same original domain may have different distributions. From Figure 7(a) and 7(b), we can see that our domain splits are better. Moreover, to eliminate the influence of class-distribution within each domain, we utilize *an adversarial way* to generate *class-invariant features* with *accurate class labels*.
>
> 2. Missing references.
>
> Yes, we miss these two excellent papers ([1][2]) for time-series domain generalization and we add these two citations in the revised version (Sec. 4 in the main paper). Additionally, we will replace 'under explored' with 'less studied' in the paper for a more accurate description. We have paid attention to some time-series domain generalization methods, e.g. AdaRNN and GILE. The detailed comparison results are shown later. Specifically, DFDG [1] aligns class relationships of samples through class-conditional soft labels, and uses saliency maps, traditionally developed for post-hoc analysis of image classification networks, to remove superficial observations from training inputs. CCDG [2] is a conditional contrastive domain generalization approach for fault diagnosis of rolling machinery and utilizes mutual information to learn domain-independent class representation. Our method pays attention to exploiting diverse sub-domains and learning generalized representations in an adversarial way. **Detailed comparisons are in the last table of this comment.**
>
> 3. Novelty on pseudo domain labels.
>
> Pseudo labeling is a general framework in semi-supervised learning and the key here is *how to utilize it to solve our problem.* Note that, we are not simply applications of pseudo labeling, but more importantly, we **propose pseudo domain-class**, **adapted class-invariant feature learning**, and **adapted** domain-invariant feature learning for fine-grained feature, pseudo label generation, and classification respectively. Moreover, we also provide theoretical insight for more natural motivations. These are all **different** from existing work. For example, [3] is a method with clustering using specific image features and it is specifically designed for CV. In addition, we have compared our method with some methods which do not require domain labels, e.g. ANDMask, AdaRNN, and RSC.
>
> 4.  Experiments setting on domain splits.
>
> There might exist some *misunderstandings* of the experiments. Since some SOTA compared methods require domain labels, we split data according to some factors, e.g. person. These splits can be inaccurate. In contrast, our method does NOT rely on artificial splits and learns splits by itself. Therefore, ours can achieve better performance and can be more universal.
>
> 5. Missing Time series domain generalization baselines.
>
> We have included some latest time-series domain generalization baselines, e.g. AdaRNN and GILE. To further eliminate your concerns, we add more comparisons on EMG and DSADS with the work that the reviewer mentioned (DFDG and CCDG).
>
>
> |           |  EMG  |       |       |       |       |
> |-----------|:-----:|:-----:|:-----:|:-----:|:-----:|
> | Target    | 0     | 1     | 2     | 3     | AVG   |
> | ERM       | 62.6  | 69.9  | 67.9  | 69.3  | 67.4  |
> | RSC       | 70.1  | 74.6  | 72.4  | 71.9  | 72.2  |
> | DFDG      | 69.5  | 72.3  | 72.6  | 70.2  | 71.2  |
> | CCDG      | 70.8  | 76.2  | 73.1  | 72.0  | 73.0  |
> | DIVERSIFY | 71.7  | 82.4  | 76.9  | 77.3  | 77.1  |
> |           | DSADS |       |       |       |       |
> | Target    | 0     | 1     | 2     | 3     | AVG   |
> | ERM       | 83.1  | 79.3  | 87.8  | 71.0  | 80.3  |
> | GILE      | 81.0  | 75.0  | 77.0  | 66.0  | 74.7  |
> | AdaRNN    | 80.9  | 75.5  | 90.2  | 75.5  | 80.5  |
> | DFDG      | 84.1  | 81.6  | 85.6  | 75.9  | 81.8  |
> | CCDG      | 86.2  | 83.7  | 87.9  | 79.2  | 84.3  |
> | DIVERSIFY | 90.4  | 86.5  | 90.0  | 86.1  | 88.2  |
>
> If you think these responses address your concerns, please consider increasing your score. Thank you!

---

> > ### Comment · Reviewer_SWyN · 2022-12-04
> > **Thanks for your response**
> >
> > Thanks for your response, it addressed most of my concerns. I would like to increase my score.

---

> > > ### Author Response · Authors · 2022-12-04
> > > **Thanks for your acknowledgment.**
> > >
> > > Thank you for recognizing our work and rebuttal. If you still have any other questions, please don't hesitate to contact us.

---

### Author Response · Authors · 2022-11-11
**Response to all reviewers**

We sincerely thank all reviewers for their time and suggestions. We are glad that all reviewers find our work 'significant and important', 'a new perspective', and 'theorical insights' (Reviewer SWyN), 'develop the algorithm and modeling end to end' and 'extensive real-data experiments' (Reviewer x6zY), 'clearly written and easy to follow', 'The design is sensible', and 'insightful' (Reviewer oHGw), 'Strong results', 'Extensive experimentation', and 'Intuitive explanation' (Reviewer EJKA), 'Improved performance' (Reviewer MfQf). We revised the paper according to the reviewers' suggestions. The newer version of the paper is uploaded. Moreover, we upload our *codes* as part of the supplementary.

We sincerely our response can address all you concerns:) If you have any questions, please let us know:)

---

### Author Response · Authors · 2022-11-17
**Reminder of discussion**

Dear Reviewers, AC, and PC,

We sincerely thank you for your time and suggestions. We tried our best to address your concerns. There are only less than **THREE** days left for discussion. Could you please read the rebuttal and respond to us? If there still exist some other concerns, please let us know.

Thanks a lot!

---

### Author Response · Authors · 2022-11-18
**General response**

Dear reviewers,

We are glad that reviewer MfQf responded to our feedback. As a summary, we are grateful that merits in novelty, methodology, theory, and empirical results are favored by you:
- Reviewer SWyN: find our work 'significant and important', 'a new perspective', and 'theorical insights'
- Reviewer x6zY: 'develop the algorithm and modeling end to end' and 'extensive real-data experiments'
- Reviewer oHGw: 'clearly written and easy to follow', 'The design is sensible', and 'insightful'
- Reviewer EJKA: 'Strong results', 'Extensive experimentation', and 'Intuitive explanation'
- Reviewer MfQf: 'Improved performance'

While we tried hard to help MfQf better understand our work and contributions, we are open to more concerns and questions of other reviewers. Thank you for your help in making this work stronger:)

---

### Author Response · Authors · 2022-12-05
**Responses are appreciated**

Dear all,

We would like to thank reviewer MfQf and SWyN for your response to our rebuttal. And we are waiting for reviewers *x6zY, oHGw, and EJKA* for your responses. If you have any other comments, please let us know.

Thanks

Paper2194 authors

---

### Decision · Program_Chairs · 2023-01-20

**Decision:**

Accept: poster

**Justification For Why Not Higher Score:**

The average score of this paper is not high (5.8). Even if we don't consider reviewer MfQf (his confidence is low), the average score is 6, indicating reviewers still have some concerns. Thus I would like to recommend Accept with poster, instead of oral or spotlight.

**Justification For Why Not Lower Score:**

I would like to recommend Accept with poster for this paper for the strengths mentioned above. 1. Time-series domain generalization is an important problem. 2. Even though some reviewers have concerns about the novelty of the method, I think it is well adapted from existing methods and has some novelty. 3. Experimental results are convincing.

**Metareview: Summary, Strengths And Weaknesses:**

The authors proposed a domain generalization method for time series representation learning. They aimed to improve the generalization on unseen data by identifying sub-distributions inside the data. They also learned domain invariant features among predicted domains and sub-domains via adversarial training. Experimental results on four public time series data show the effectiveness of the proposed method.

Strengths:
1. Time-series domain generalization is a new and important research topic.
2. The experiments are quite comprehensive. After the rebuttal, it is very nice that the authors included additional baselines and backbone networks as suggested by the reviewers.

Weaknesses:
1. In the proposed framework, existing methods are used in different components. For example, DANN together with gradient reversal layer (GRL) is used for domain-invariant representation learning. Therefore, some reviewers have the concern about the novelty of the proposed method.
2. A reviewer (MfQf) thinks that the writing of this paper is not clear, and many technical details are missing.


**Note From Pc:**

if the above contains the word "oral" or "spotlight" please see: "oral" presentation means -> notable-top-5% and "spotlight" means -> notable-top-25%. As stated in our emails, we are disassociating presentation type from AC recommendations

**Summary Of Ac-Reviewer Meeting:**

A virtual meeting with Reviewers oHGw and SWyN was held on 10 Dec. Reviewer MfQf did not join the meeting as he was on travel, while he told me via email that he will not change the rating. Other two reviewers did not join the meeting.

Reviewer oHGw still had the concern about the novelty and he would not change his rating.

Reviewer SWyN shared that his concern about comparison with recent baselines had already been addressed. However, he thought the novelty of the method can be further improved, even though the problem is new. As such, he would also not change his rating.